# Self-assembly of Co/Pt stripes with current-induced domain wall motion towards 3D racetrack devices

Pavel Fedorov[1,2] ✉, Ivan Soldatov [2], Volker Neu[2], Rudolf Schäfer [2,3], Oliver G. Schmidt [1,4,5] ✉ & Daniil Karnaushenko [1] ✉

Modification of the magnetic properties under the induced strain and curvature is a promising avenue to build three-dimensional magnetic devices, based on the domain wall motion. So far, most of the studies with 3D magnetic structures were performed in the helixes and nanowires, mainly with stationary domain walls. In this study, we demonstrate the impact of 3D geometry, strain and curvature on the current-induced domain wall motion and spin-orbital torque efficiency in the heterostructure, realized via a self-assembly rolling technique on a polymeric platform. We introduce a complete 3D memory unit with write, read and store functionality, all based on the field-free domain wall motion. Additionally, we conducted a comparative analysis between 2D and 3D structures, particularly addressing the influence of heat during the electric current pulse sequences. Finally, we demonstrated a remarkable increase of 30% in spin-torque efficiency in 3D configuration.

In an effort to increase data storage density, various strategies are being explored to move from 2D to 3D electronics designs. However, implementing such a transition for wafer-scale production using current micro-fabrication technologies and materials is exceedingly challenging. Following our previous research demonstrating wafer-scale fabrication of self-assembled 3D devices[1–6] this work demonstrates roll-down self-assembly of magnetic memory unit, based on the racetrack (RT) architecture. The approach leverages a self-assembled shapeable polymeric platform and standard microfabrication technology. The flexible platform allows the shaping of thin metal wires into new forms, facilitating the construction of different configurations of magnetic devices and their characterization. Transforming flat ferromagnetic stripes into the ring or spiral geometry induces strain that influences the magnetic properties, for example, perpendicular magnetic anisotropy (PMA), spin-orbit torque (SOT), and domain wall (DW) dynamics[7–9]. Strain-induced effects have been examined in films with perpendicular and in-plane magnetic anisotropy using piezoelectric-controlled stages[10–15] and self-assembled, shapeable polymeric material systems[16–18]. These

research papers showed that strain and curvature-induced effects have a strong influence on the thin film's magnetic properties, however, the curvature contribution becomes pronounced at small radii (order of hundreds of nanometers)[19–21].

Recent reports have delved into several avenues to explore the DW motion in 3D geometries[22–24]. The propagation of DWs in curved nanowire was theoretically studied in cylindrical magnetic nanowires, revealing non-trivial oscillatory behavior, that depends on the curvature of the wire[25,26]. Several research groups presented studies on DWs and magnetic textures in 3D ferromagnetic nanohelices[22,27,28] fabricated with focused electron beam-induced deposition. A recent report[23] further showcased the realization of current-induced domain wall motion in 3D racetrack (RT) based on synthetic antiferromagnets with PMA, fabricated on a free-standing tilted membrane. These domains can be efficiently manipulated through the application of ultra-short current pulses[29,30]. The RT stands out for its immense potential to achieve significantly higher data density storage compared to other emerging memory technologies. A crucial aspect of the RT is its ability to reliably

[1]Research Center for Materials, Architectures and Integration of Nanomembranes (MAIN), Chemnitz University of Technology, 09126 Chemnitz, Germany. [2]Leibniz Institute for Solid State and Materials Research, 01069 Dresden, Germany. [3]Institute for Materials Science, TU Dresden, 01062 Dresden, Germany. [4]Material Systems for Nanoelectronics, Chemnitz University of Technology, 09107 Chemnitz, Germany. [5]Nanophysics, Faculty of Physics, TU Dresden, 01062 Dresden, Germany. ✉e-mail: pavel.fedorov@main.tu-chemnitz.de; oliver.schmidt@main.tu-chemnitz.de; daniil.karnaushenko@main.tu-chemnitz.de

manipulate data at high speed while consuming low power[31,32]. This unique combination of features positions the RT as a leading candidate for the realization of next-generation memory devices, offering exceptional performance and non-volatility.

One promising approach that fulfils these criteria is the current-induced domain wall motion (CIDWM) method, enabling the local manipulation of magnetic domains in a field-free manner. Initial research on CIDWM was focused on micro-scale stripes with in-plane magnetic anisotropy, where domain walls (DW) were propelled by spin-polarized current that exerts a spin transfer torque (STT)[30,33] on the walls. Later the interest has shifted towards stripes with perpendicular magnetic anisotropy (PMA), as domain walls in such structures exhibit higher velocities and improved thermal stability[34-38]. More recent research shows great potential for enhancing the performance and efficiency of RTM-based memory devices, which can particularly be achieved by engineering the strain in magnetic thin films.

Current-induced domain wall motion in heavy metal/ferromagnet (HM/FM) bilayers makes it possible to achieve high velocities at relatively low current densities[32,39]. This phenomenon arises due to the combined effects of spin-orbit torque[40,41] (SOT) and Dzyaloshinskii-Moriya interaction (DMI)[32,42-44]. A SOT is generated by the spin Hall effect from the HM layer, which is maximum for a Neel-type DW, while DMI emerges at the interface between the HM and FM layers. In ultrathin ferromagnetic films with PMA, the DMI transforms the Bloch-type domain wall into a chiral Neel-type domain wall[42]. This results in chiral DWs with a magnetization rotation perpendicular to the DW plane. Further research has shown that in a three-layer system consisting of HM/FM/HM[45-47], where the two HM layers have opposite spin Hall angles (SHA), the domain wall propagates at a lower current density of approximately $-10^{10}$ A/m². The generated spin current from the two HM layers increases the total SOT on the domain wall, thereby reducing the critical current. CIDWM has also been observed in synthetic antiferromagnets with PMA, achieving a speed of 750 m s$^{-1}$, attributed to the presence of Neel-type domain walls and nearly zero net magnetization[48,49]. In another study[50], a low current density CIDWM ($\sim 10^9$ A m$^{-2}$) was demonstrated in the magnetic semiconductor GaMnAs. However, this result was achieved at a temperature of 100 K due to the semiconductor's low Curie temperature. Furthermore, it was found that domain walls in ferrimagnets exhibit high velocities at the angular momentum compensation temperature, where they undergo a transition from a ferromagnetic (FM) to an antiferromagnetic (AFM) state while maintaining a nonzero net magnetic moment, enabling a reliable detection[51-54]. The RT architecture was proposed for quantum computing, utilizing DWs as flying qubits[55]. The manipulation and transfer of information stored in the chirality of the DWs can be achieved by shuttling the qubits along the racetrack.

Figure 1a, b show schematically the layout of the write-, read-, and shift functionalities in the fabricated devices. The RT line itself has dimensions of 3 μm width and 100 μm length. The fabrication process employed e-beam lithography (EBL) and lift-off techniques. The film stack of Ta$^{3.0\,nm}$/Pt$^{5.0\,nm}$/Co$^{0.7\,nm}$/Ta$^{4.0\,nm}$/Pt$^{1.0\,nm}$ was deposited by magnetron sputtering at room temperature. In a subsequent lithography step, Cr$^{10.0\,nm}$/Au$^{50.0\,nm}$ contact pads and Ta$^{5\,nm}$/Cu$^{250\,nm}$/Ta$^{10\,nm}$/Cr$^{5\,nm}$/Au$^{40\,nm}$ injection line were manufactured. An injection line is used to locally create a magnetic domain in the FM strip; a Hall cross is used to detect the magnetization of the shifted DWs. Finally, the planar structure was successfully transformed into 3D by the strain-induced rolling process, as illustrated in Fig. 1 c (Materials and Methods section). Figure 1d displays an optical micrograph of the 3D device post self-assembly, showcasing a polymer tube with a diameter of Ø32 μm with an induced strain in the magnetic stripe equal to 1.31% (supp material). The developed self-assembly process exhibits a high yield, commonly up to 90%[1]. The final diameter of the tube is not affected by the functional thin-film structures, i.e. the gold contacts and the magnetic stack, as their thicknesses are significantly below that of the PI layer (600 nm).

Magneto-optical Kerr microscopy was utilized to conduct a comprehensive analysis of the CIDWM in both, the 2D and 3D devices, which obtain their performance through the combined effects of SOT and DMI in the Pt/Co/Ta stack. We discovered that the device's shape significantly influences the speed of the domain wall in the 3D configuration. Additionally, we thoroughly examined the impact of different substrates and state conditions on the device performance, including SOT, DMI, and Joule heat effects.

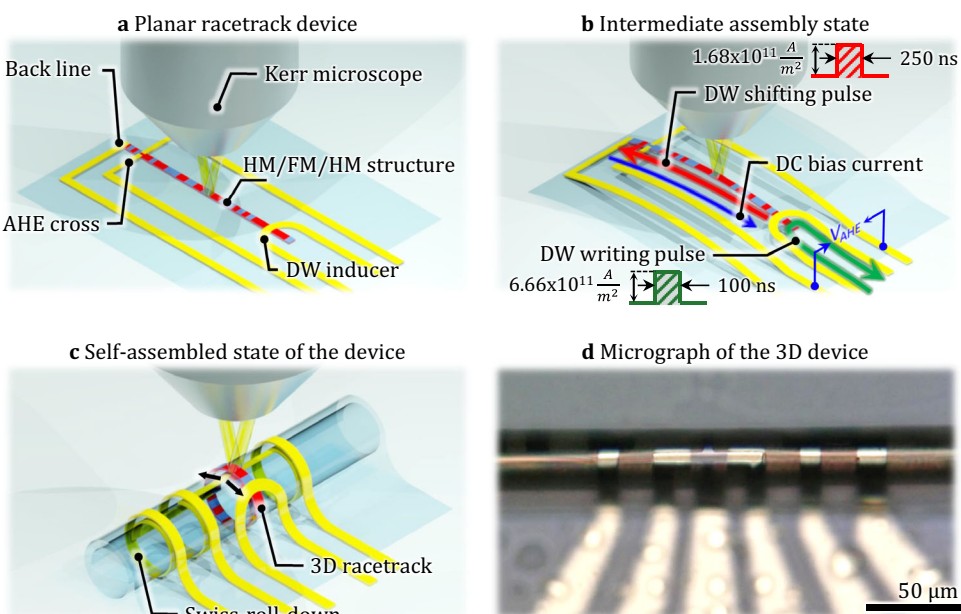

**a** Planar racetrack device

Back line — Kerr microscope
AHE cross
HM/FM/HM structure
DW inducer

**b** Intermediate assembly state

$1.68\times10^{11}\frac{A}{m^2}$ — 250 ns
DW shifting pulse
DC bias current
DW writing pulse — $V_{AHE}$
$6.66\times10^{11}\frac{A}{m^2}$ — 100 ns

**c** Self-assembled state of the device

3D racetrack
Swiss-roll-down

**d** Micrograph of the 3D device

50 μm

**Fig. 1 | 3D rolled-down racetrack memory device.** Schematic illustration of **a** 2D RT on the top of a shapeable platform; **b** intermediate state of the self-assembling, rolling-down approach and **c** rolled-down 3D device after self-assembling into the "Swiss-roll". Black arrows indicate tensile strain in the magnetic layer. **d** Optical image of the 3D self-assembled device, with a final diameter of 30.9 μm.

## Results

### Material characterization

The 3D RT self-assembly is achieved using a shapeable polymeric platform composed of three functional layers: a sacrificial layer (SL), a polyimide layer (PI), and a hydrogel layer (HG)[31,44]. The polymeric films are spin-coated and patterned using direct UV lithography. The top HG layer has a surface cut in the bottom middle area, providing access to the PI layer where the functional magnetic layers are structured, see Fig. 2a, b. The growth of the film stack HM/FM/HM is subsequently accomplished on top of the shapeable polymeric layers forming planar devices. For reference, the same layer stacks have been prepared directly on the surface of a bare glass substrate to compare the device performance, see Fig. 3a, b.

We employed the polar magneto-optical Kerr effect (PMOKE) technique with out-of-plane sensitivity to measure the coercivity ($H_c$) of both the extended film and the racetrack micro-stripe, see Fig. 3a, b. The observed square loop (Fig. 3c, d) reveals the presence of perpendicular magnetic anisotropy (PMA) in all samples. Specifically, we observed $H_c$ values of 11.2 mT and 16.6 mT on the glass (Fig. 3c) and 10.9 mT and 11.8 mT on the PI substrate (Fig. 3d) for the film and stripe, respectively. The field sweep speed was set at 0.5 mT/s. While the coercivity of the extended films shows a similarity for both substrates, we noted a significant discrepancy in the case of the patterned stripes. Subsequently, we conducted AFM measurements on the fabricated samples to examine the surface roughness. Our findings revealed that the glass substrate exhibits a flatter surface but possesses a larger roughness compared to the PI substrate. The roughness $R_{rms}$ of the substrates in the vicinity of the RT (red squares in Fig. 3e and f) were estimated to be 1.75 nm for glass and 0.78 nm for PI. To minimize potential contributions from surface waviness, the AFM scans were performed in an area with minimal height variation.

The wavy surface of the PI layer is a consequence of a rapid thermal shrinking/expansion during the fabrication process of the polymeric platform. Notably, during the cooling step, the glass

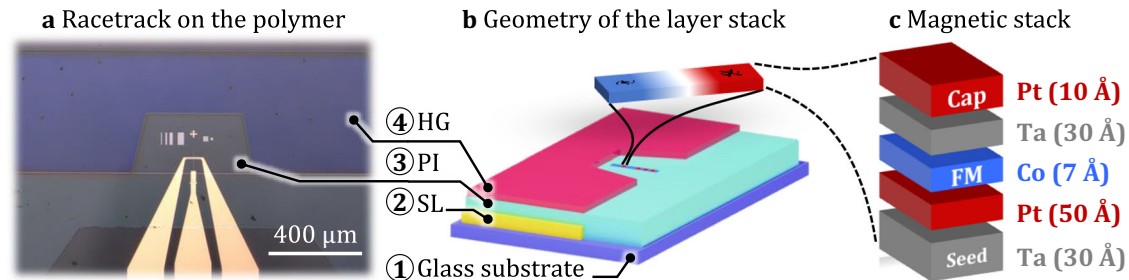

**a** Racetrack on the polymer          **b** Geometry of the layer stack          **c** Magnetic stack

④ HG
③ PI
② SL
① Glass substrate

400 µm

Cap — Pt (10 Å)
Ta (30 Å)
FM — Co (7 Å)
Pt (50 Å)
Seed — Ta (30 Å)

**Fig. 2 | Geometry and the device layer stack. a** Optical image of the RT device on the polyimide before the self-assembly process. The reinforcing PI layer has a dimension of 0.5 × 1.5 mm. **b** Geometry and composition of the polymeric layer stack. **c** Exploded view of the magnetic layer stack.

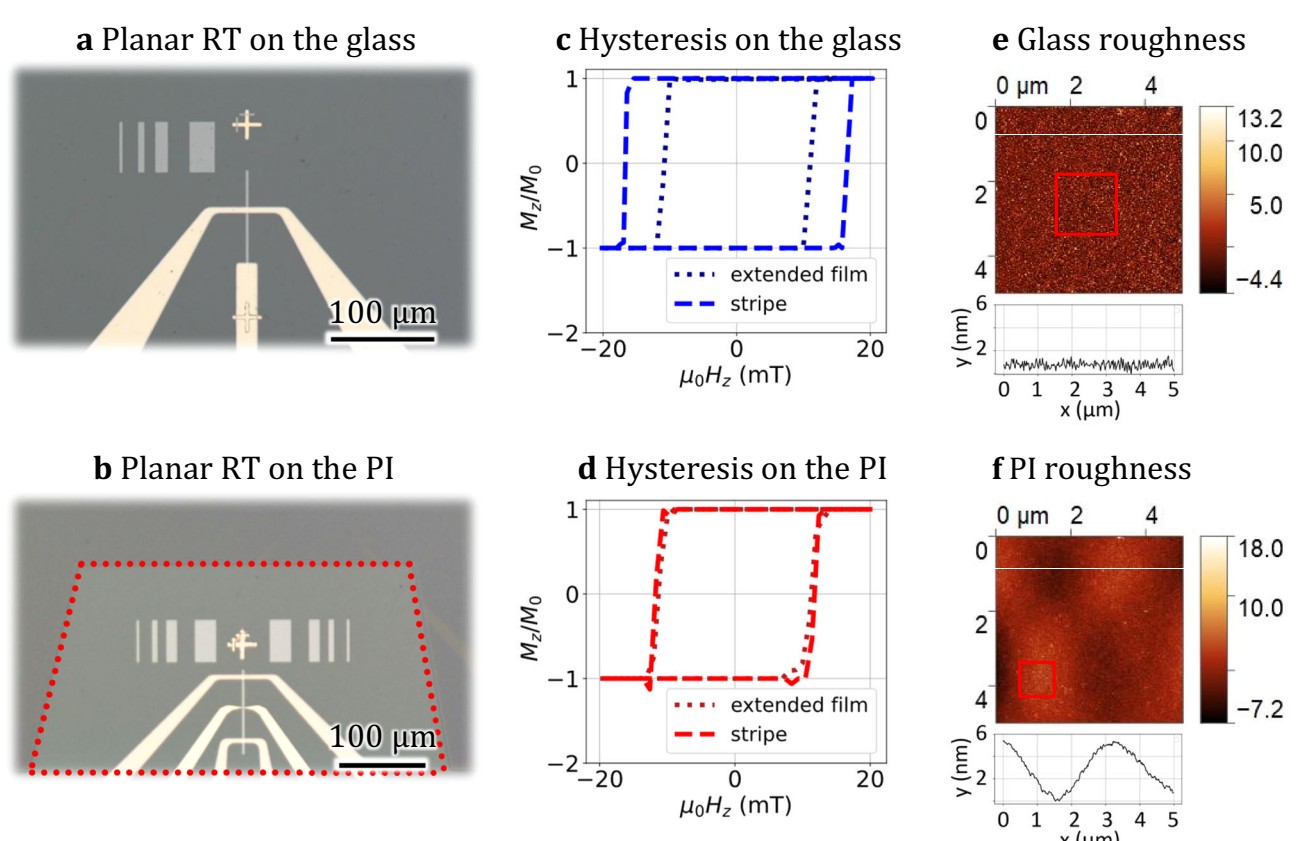

**a** Planar RT on the glass          **c** Hysteresis on the glass          **e** Glass roughness

100 µm

····· extended film
- - - stripe

**b** Planar RT on the PI          **d** Hysteresis on the PI          **f** PI roughness

100 µm

····· extended film
- - - stripe

**Fig. 3 | Planar racetrack devices. a** Optical image of the RT device on the glass **b** and the PI layer with Hall contacts. **c** Hysteresis loops of stripe and extended film deposited on the glass **d** and polymer. **e** AFM scans and a single scan line showing the roughness of the glass **f** and polymeric surfaces.

substrate undergoes a faster lateral size reduction compared to the polymers, leading to the formation of micrometer-scale wrinkles on the surface of the polymer layer. This observation highlights the crucial importance of carefully considering the substrate properties during the fabrication process.

## Domain wall motion in 2D planar and 3D Swiss roll-down geometries

To detect and record the motion of DWs in our 3 µm wide and 100 µm long stripes, we utilized magneto-optical Kerr microscopy[56] and Anomalous Hall Effect (AHE) measurements. The DW injection was achieved by applying a local Oersted field, generated by a current pulse through the thick DW inducer strip line as shown in Fig. 1b. To drive the DW itself, a train of pulses with a pulse width of 250 ns and a period of 1 ms was applied through the magnetic stripe (Fig. 1b), without the need for an externally assisting magnetic field. This pulse period was chosen to prevent long-term Joule heat accumulation within the structure. The DW velocity was determined by dividing its total propagation distance by the total current ON time during the pulse train. We maintained a distance of 70 µm between the DW inducer and the back line, being long enough to keep the observed DW propagation during the pulse train within the field of view in the microscope, thus not requiring a frequent resetting of the magnetic state of the wire. For statistical purposes, we repeated the experiment ten times for each current value.

The Hall cross electrodes (Figs. 3b and 4a, bottom) with a width of 300 nm were patterned on the top of the RT stripe to detect the DW position. The thin electrodes were made of Cr$^{10\,nm}$/Au$^{50\,nm}$ to prevent DW pinning and maintain enough flexibility to prevent de-attachment from the surface during the rolling process. Simultaneously with the optical observation, while a series of current pulses were applied to drive the DW, the Hall voltage $V_{Hall}$ was measured after each pulse train to check the presence of a domain wall. A clear $V_{Hall}$ drop, see Fig. 4a, b, indicates that a DW arrives at the Hall cross area and experiences pinning before further propagation. This pinning is due to the higher conductivity in the area of the Hall electrodes, which reduces current density required for shifting the DW.

Figure 5a presents typical Kerr images of CIDWM in the 2D device on the glass, in response to a series of injected 250 ns long current pulses, compose of 100 pulses. Dark region corresponds to magnetization in -$Z$ direction. The DW propagates along the current direction.

For the 3D racetrack, a Ø32 µm tube has been achieved after the tube rolling process where the RT device occupies the tube circumference, as depicted in Fig. 1c. The DW speed in the 3D racetrack was measured using Kerr microscopy, with a focus on a specific area located on the dome of the tube within the microscope's field of view. Typical Kerr images of CIDW propagation were obtained (compare Fig. 5b), and the results for the 2D and 3D racetrack configurations are compared in Fig. 5c, d.

Figure 5c illustrates the dependence of the DW velocity on the current density for both 2D and 3D RT configurations, with the DW propagating along the direction of the current flow[41,43,48]. The minimum current densities, required for the depinning of the DWs, were found to be 1.34 x 10$^{11}$ A m$^{-2}$ and 1.68 x 10$^{11}$ A m$^{-2}$ for 2D and 3D configurations, respectively. As the current density increases, the velocity exhibits an exponential increase, corresponding to the creep regime of the DW motion[39,57,58]. The maximum current densities for 2D and 3D samples were 2.43 x 10$^{11}$ A m$^{-2}$ and 1.97 x 10$^{11}$ A m$^{-2}$, respectively. Beyond these values, random domain nucleation occurred due to Joule heating. The field-free motion of the DW along the current direction is attributed to the physical mechanisms described in the introduction, namely the presence of Dzyaloshinskii-Moriya Interaction (DMI) and SOT in the Pt/Co/Ta stack, as reported in[45,59]. DMI occurs at the Co/Pt interface due to spin-orbit interaction, stabilizing a left-handed chiral Neel-type DW. Simultaneously, the induced spin current influences the DW magnetization, resulting in a DW uniform propagation.

The mean DW velocity in the creep regime can be described as[58]:

$$v = v_0 \exp\left[-\frac{U_c}{k_B T}\left(\frac{H}{H_{dep}}\right)^{-\mu}\right] \quad (1)$$

where $v_0$ is the threshold velocity, $k_b$ the Boltzmann constant, U$_c$ is the pinning potential, H$_{dep}$ is a critical depinning field ($\equiv$j the depinning current density), µ is a dynamic exponent that is equal to -¼ for a 1D interface moving in the 2D weakly disordered medium. To check whether the DW velocity roughly obeys this law, we plot ln($v$) versus j$^{-0.25}$ ($\equiv H^{-0.25}$) in Fig. 5d. A linear dependency is valid for the whole operation range for the 3D RT; for the 2D state, it is valid below a current density of 1.82 x 10$^{11}$ A m$^{-2}$. We observe a deviation from the linear trend at higher values, which is consistent with the system entering into the depinning and flow regime of operation, as it was reported previously[39].

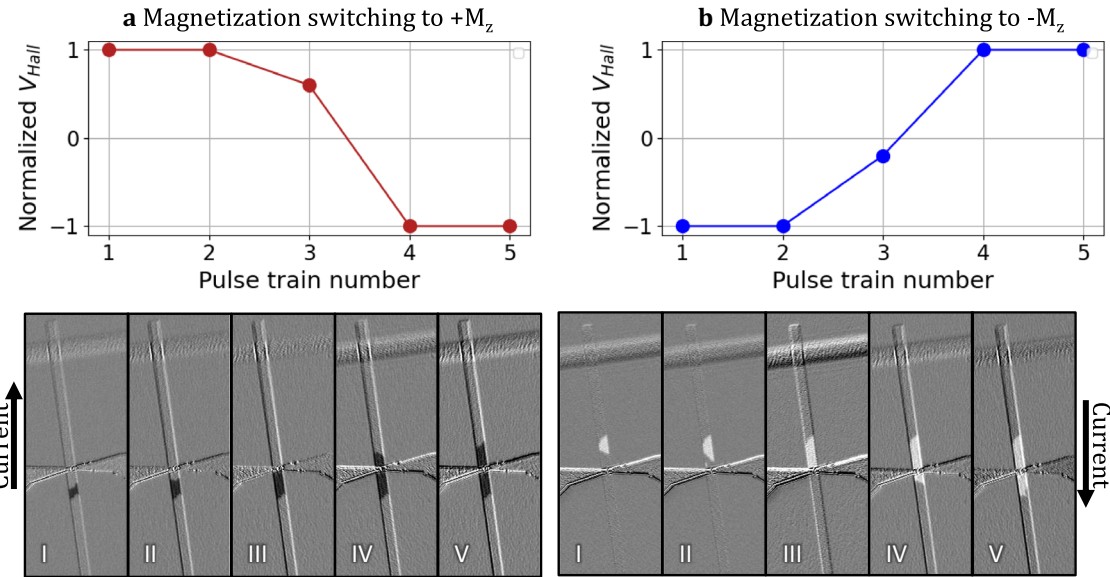

**a** Magnetization switching to +M$_z$

**b** Magnetization switching to -M$_z$

**Fig. 4 | Current induced DW propagation and detection.** Direct observation of propagation and detection of **a** up-down and **b** down-up DW at the Hall cross. The metallic Hall probe electrodes create a pinning site along the wire.

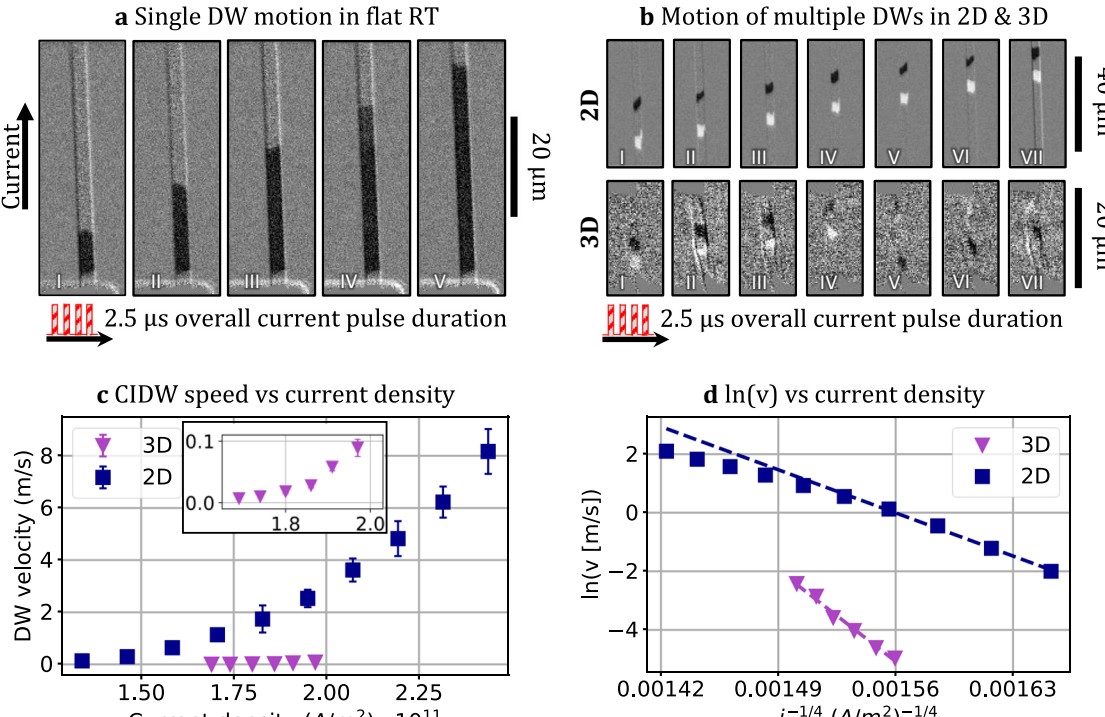

**Fig. 5 | Domain wall propagation in 2D and 3D racetracks. a** Kerr images of a current-induced domain wall motion in the 2D racetrack on a glass substrate. **b** Kerr images of current-induced motion of multiple domain walls and domains in 2D (upper) and 3D (bottom) racetracks. **c** Comparison of the DW velocities for the 2D and 3D cases, driven by current pulses. The inset shows the zoom of the DW velocity for the 3D case. **d** DW velocities versus current density $\ln(v) - j^{-0.25}$, plotted with the creep law fitting for the 2D and 3D cases.

We have observed a reduction in the DW velocity in the 3D race-track device, which is rather counterintuitive. We have anticipated that the thermal contribution would aid in DW propagation, resulting in a higher speed in the rolled state. The decrease in the DW speed can be associated with the PMA increase[14,15] and the formation of nano-scale cracks in the RT stripe[18] due to the tensile in-plane strain. The increase of the PMA occurs due to the magnetostrictive properties of the Pt/Co, in response to the applied tensile strain. The coercivity of the magnetic stripe increased by 17.5% under the applied strain, due to the geometry change (Supp material, Fig. 3). Therefore, we assume that the PMA of the Pt/Co/Ta stack increases as well. The quantitative change of the PMA cannot be deduced from this number, however, and the standard method of the anisotropy field measurement cannot be applied in the current 3D configuration. The change of PMA leads to a change of the Bloch domain wall energy $\gamma = \pi\sqrt{AK_{eff}}$, with $A$ being the exchange stiffness and $K_{eff}$ the magnetic anisotropy constant. This change is equivalent to the growth of a creep energy barrier in the system. According to the creep law and the exponential speed dependence on pinning potential, a small change in the energy barrier leads to a larger velocity change that hinders a fast DW propagation.

Another factor that could influence on the DW speed reduction is the formation of micro- and nano-cracks[18] in the RT stripe. A thin metallic film experiences a linear elastic deformation with minor resistance variation under an elongation strain below 5%. In our work, the 3D racetrack experiences only 1.31% of strain, which seems to be sufficient to form defects in the RT that result in additional pinning centers.

Consequently, there are several opposing effects that determine the final DW speed: the SOT efficiency, the PMA of the multilayer, and the formation of defects. Under applied in-plane elongation strain, both the SOT efficiency (see the next section) and the PMA of the FM layer increase[15]. Eventually, the DW speed in a 3D rolled-down configuration depends on the "weight" of each of the contributions. In the reported case, the effect of the PMA increases and the formation of strain-induced defects prevails over the SOT, leading to the lowering of the DW velocity. However, there are ways to enhance the speed, for example: to change the rolling direction without changing the HM/FM thickness, or increasing the FM layer thickness to the point where the anisotropy transition occurs. The increase of the HM thickness (above 5 nm) will not give much contribution, due to the short spin diffusion length of 3 nm.

**Spin-orbital torque analysis**

In this section, we examine the impact of applied strain and changes in geometry on the SOT in perpendicular magnetic anisotropy structures using a self-assembly approach. To conduct this study, we explored a rolling-down configuration (Fig. 1c) that induces tensile strain on the magnetic stack structures, black arrows. By performing systematic measurements, we aimed to assess the effect of shape-induced strain on the SOT efficiency of the Pt/Co/Ta multilayer. To achieve this, we adopted the current-induced hysteresis loop shift method[45,59], which offers a simple measurement configuration, see Fig. 6a. This method allows for the simultaneous estimation of the SOT efficiency and the DMI field. The SOT efficiency was measured by patterned Hall bar structures, as sketched in Fig. 6a. These structures were 5 μm wide and 60 μm long. The devices were prepared using optical lithography and the Ar-ion milling technique. A second lithography step was performed to deposit $Cr^{10.0\,nm}/Au^{50.0\,nm}$ contact pads. The SOT measurements were conducted for two configurations: first, in a planar configuration with structures fabricated on a 1 mm thick glass substrate, and second, in a rolled-down polymer tube configuration.

The current-induced shift of the AHE hysteresis loops was measured as a function of DC current in the presence of an in-plane magnetic field $H_x$. Figure 6 a represents the experimental configuration for the planar case. The current passing through the wire generates an effective field $H_{SHE}$ in an out-of-plane direction, attributed to the

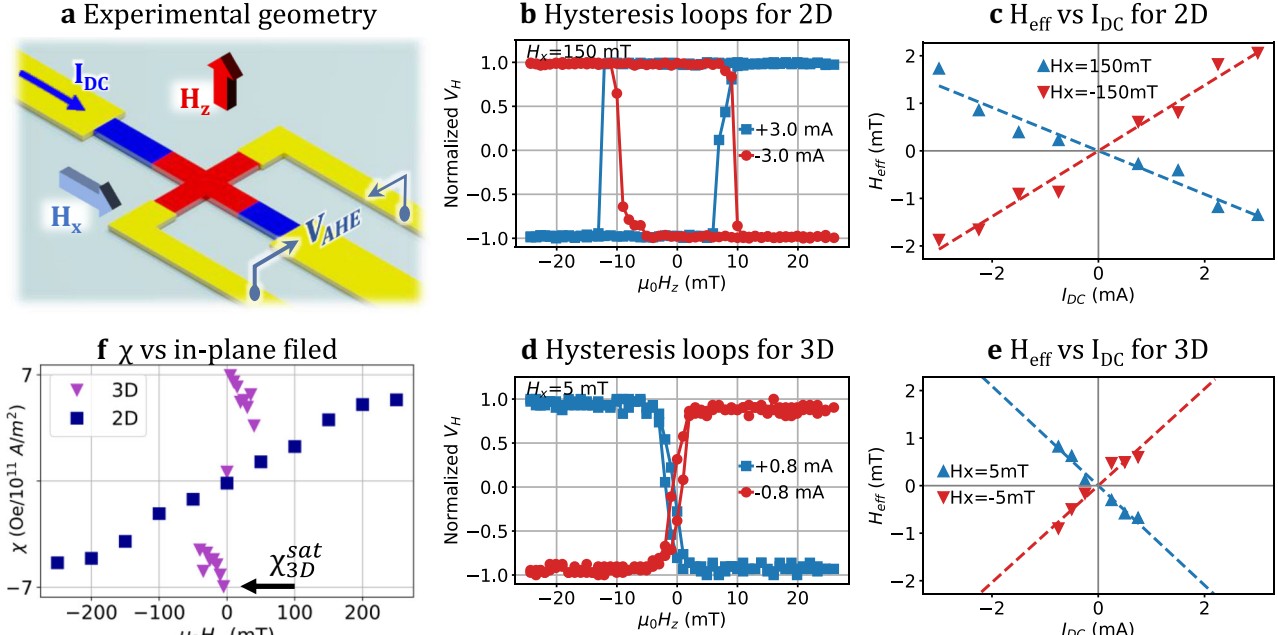

**Fig. 6 | SOT efficiency using current-induced hysteresis loop shift method for Ta$^{3.0 nm}$/Pt$^{5.0 nm}$/Co$^{0.7 nm}$/Ta$^{4.0 nm}$/Pt$^{1.0 nm}$ stack. a** Schematics of the Hall bar and the measurement configuration. **b** Typical anomalous Hall loops for planar stack fabricated directly on the glass substrate. **c** $H_{eff}$ as a function of $I_{DC}$ under different bias fields $H_x$ for the planar case. **d** Typical anomalous Hall loops for a 3D curved stack fabricated on a polymeric self-assembled rolled-down strained architecture. **e** $H_{eff}$ as a function of $I_{DC}$ under different bias fields $H_x$ for the 3D strained case. **f** SOT efficiency $\chi$ as a function of the in-plane field for two substrates.

Slonczewski-like torque, which influences the magnetization dynamics of the domain wall (DW), and equals to[45,60]: $H_{SHE} = \frac{\hbar\theta_{SHE}j}{2eM_s t_F}$ with, $\theta_{SHE}$ – effective spin Hall angle, $j$- current density, $M_s$- saturation magnetization, $t_F$ – ferromagnetic layer thickness. In the absence of an external in-plane field $H_x$, the DMI in Pt/Co/Ta stabilizes a chiral Neel-type[59] DW with the magnetization $\bar{m}$ aligned along the $x$ direction.

When a DC current is applied, it generates an effective out-of-plane field $H_{eff}$, which induces the unidirectional motion of domain walls (DWs) with both chiralities, see Fig. 7a. Here, $H_{eff}$ can be expressed as $H_{eff} = H_{SHE} * \cos(\varphi)$, where $\varphi$ represents the angle between the DW moment and the $x$-axis. The application of an in-plane bias field causes the reorientation of moments in Néel walls towards the $H_x$ direction. If the strength of the in-plane field $H_x$ exceeds the DMI field, it causes the magnetic moments to align uniformly in one direction for all DWs in the stripe, see Fig. 7b. Hence, $H_{eff}$ will point in one direction for both walls, resulting in parallel or antiparallel alignment with the $H_z$ field during the out-of-plane field-driven magnetization switching, thus leading to a shift in the hysteresis loop.

Figure 6b displays the typical Anomalous Hall Effect (AHE) loops for the planar sample with a superimposed in-plane field of 150 mT for opposite currents. The observed shift in the hysteresis loops (with field sweep along $H_z$) is attributed to the Slonczewski-like torque. Specifically, a positive (negative) $j_{DC}$ generates an effective positive (negative) $H_{eff}$ field that acts on the Néel-type chiral DW. However, it is important to note that the DC current in the structure also results in Joule heating ($-j^2$), which leads to a reduction of the switching field. This effect becomes particularly significant in devices constructed on rolled tubes, where the limited thermal conductivity of the polymer causes a decrease in the coercivity of the multilayer already at smaller current values. Thereby, the effective field can be expressed like: $H_{eff} = (H_{SW}^{DU} + H_{SW}^{UD})/2$, where the $H_{SW}^{DU}$ and $H_{SW}^{UD}$ is a down-to-up and up-to-down switching field, respectively, with no Joule heating contribution. $H_{eff}$ causes the horizontal shift of the hysteresis loop[45,59]. Consequently, the SOT efficiency can be determined by performing a linear fit of $H_{eff}$ as a function of current density, represented as $\chi = H_{eff}/j$ in Fig. 6c.

Next, the SOT efficiency in the 3D configuration was measured. After the rolling process, one of the Hall crosses has been precisely aligned on top of the polymer tube being parallel to the substrate surface during the self-assembly process. The accuracy of this alignment is confirmed through hysteresis loop measurements. The Hall voltage is the same for an out-of-plane field of 0 mT and 20 mT, meaning that magnetic vectors in the Hall cross area do not deviate further from the $z$-axis at high values of the magnetic field (Sup Fig. 2).

Figure 6d presents the results obtained from the hysteresis loop measurements on the 3D curved structure with a small in-plane bias field. When the in-plane field $H_x$ is lower than the coercivity of the stripe, the measurement conditions became similar to those for a planar structure. However, once the in-plane field is higher than the $H_c$ value, the re-magnetization process initiates at the sides of the curved structure rather than at the top. We also observe a rapid decay of $\chi$ as a function of $H_x$, as depicted in Fig. 6 e.

Figure 6f depicts $\chi = H_{eff}^z/j$, as a function of the longitudinal in-plane field for both, the planar and the 3D rolled cases. In the planar case, the efficiency $\chi$ shows a linear increase and eventually saturates at 211.7 mT, providing a rough estimation of the DMI field. Based on this measurement, we estimate $\chi_{SHE}$ to be approximately 5.3 mT $10^{-11}$A m$^{-2}$, that is close to values obtained in the similar system[45,46,61]. In contrast, the rolled sample demonstrates a different dependency of $\chi$. It reaches a saturation value of $\chi_{SHE}$ at around 6.9 mT $10^{-11}$ A m$^{-2}$ for lower values of the in-plane field ($H_x$ = 5 mT), with a rapid decay as the field increases. This observed behavior indicates that the applied strain of 1.31% leads to a 30.1% increase in the (SOT) efficiency in the rolled structure. As the applied strain is increased, the racetrack stripe gets narrower, consequently resulting in an increase in the metal's resistivity. According to the discussion[62], the heightened efficiency of SOT efficiency under elongation strain is attributed to extrinsic scattering. It was reported that in heavy metals, like Pt, Ta, W, Pd the intrinsic scattering typically dominates over the extrinsic. The total spin Hall conductivity can be expressed as a sum of the intrinsic and extrinsic parts. The total spin Hall conductivity, exhibits the increase in response to the applied elongation strain. In the systematic analysis, it

Schematic illustration of CIDWM without and with in-plane field

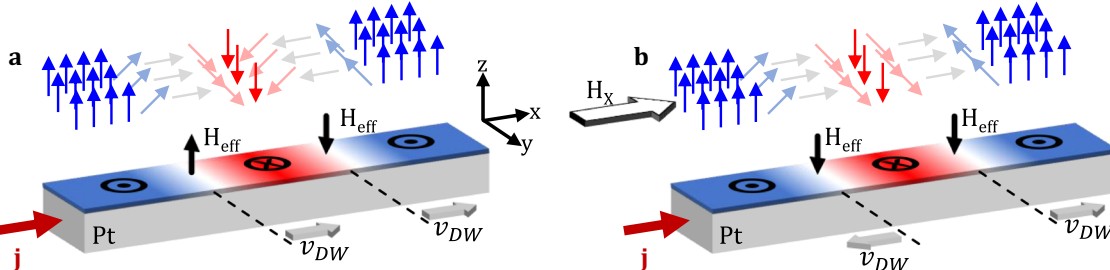

**Fig. 7 | Realignment of DWs under the influence of an in-plane field. a** Schematic illustration of CIDWM in the absence of an external in-plane field $H_x$. The effective field drives the domain walls in the same direction with respect to the current.

**b** CIDWM with in-plane magnetic field $H_x$, which realigns the domain-wall moments, resulting in a change of the $H_{eff}$ and the DW propagation direction.

was shown that the intrinsic part remains constant as function of strain. This result led to the conclusion that SOT modulation is attributed to the extrinsic part. A similar effect was previously reported in a comparable material combination (double-layer Co/Pt) with an increase of 28.5% in a Co/Pt system on a Kapton film[62].

## Joule heating contribution

To investigate the role of Joule heating on the propagation of DWs driven by current pulses, we conducted a series of time-resolved resistance measurements. To initiate our investigation, we performed a meticulous calibration of the RT resistance as a function of temperature. For accurate temperature monitoring, a temperature sensor Pt1000 was glued to the glass substrate and placed on the heater. A separate glass substrate with the 2D racetrack was mounted on the heater close to the temperature sensor, see Fig. 8a. This setup allowed us to achieve precise real-time temperature measurements. As illustrated in Fig. 8b, the resistance exhibits a linear variation with respect to the temperature, confirming the metallic nature of the device. The slope, denoted as $(\frac{dR}{dT})_{RT}$[63] = 2.8679 $\Omega$ K$^{-1}$, yields the corresponding coefficient of the change in temperature with respect to the resistance: $\gamma_{RT}$ = 0.3486 K $\Omega^{-1}$. For this measurement we applied pulses with a duration of 250 ns, with an OFF level of 0 $\mu$A and an ON level of 800 $\mu$A, to prevent undesired device heating effects.

Next, we measured the resistance change as a function of the current pulse amplitude in the 2D and 3D racetrack architectures, as described in the DW motion section. The temperature changes associated with these measurements can be estimated using the resistance variation formula: $T = T_0 + \gamma_{RT}\Delta R$, where $\Delta R = R - R_0$. Hence, we measured the resistance variation in response to different current pulse amplitudes. For this measurement, we maintained a constant pulse width of 250 ns and a selected ON high-level ON current amplitude, which was consistent with the conditions used for DW propagation. The low-level OFF current was set to 800 $\mu$A during the pulse-OFF period. Each data point was obtained from the analysis of 10 individual pulses. Using the calibration data, we estimated the temperature rise in the current pulsed mode by converting the measured resistance change into a temperature change.

Figure 8 c illustrates the characteristic temperature response of the self-assembled polymer tube when subjected to a current density of 1.79 x 10$^{11}$ A m$^{-2}$ during the pulse application. It reveals an abrupt temperature increase within nanoseconds and a further steady temperature rise throughout the 250 ns ON period, succeeded by a swift post-pulse cooling phase. It is evident that, at high current densities, the system does not achieve thermal stability within the 250 ns timeframe. Our measurements reveal that the temperature change, $\Delta T$, amounts to about 50 K in the 2D case and can reach up to 83 K for the 3D racetrack, see Fig. 8d. As anticipated, the Joule heating effect is more pronounced for the self-assembled polymer tube compared to the glass substrate. This distinction arises due to the polymer's lower

thermal conductivity of 0.12 W mK$^{-1}$ in contrast to the glass's 1.076 W mK$^{-1}$. The data points within the central region of the shaded area in Fig. 8d indicate the average temperature/resistance change observed during the pulse. The findings from this section underscore that a temperature rise of the RT in 3D, during pulse application, doesn't boost the DW energy enough to overcome the additional pinning energy arising from the strain-induced PMA modification.

## DMI characterization

The measurement of the DMI field was conducted using the asymmetric bubble expansion technique imaged with Kerr microscopy at room temperature. The efficacy of this method has been well-established in prior investigations[64,65] involving Co/Pt and Co/Pd multilayer films to assess DMI effects. The procedure involves initiating a small OOP field pulse with a fixed magnitude to nucleate and slightly expand a magnetic bubble domain (indicated by the grey-shaded region at the center, enclosed by a dashed boundary in Fig. 9a, b). A reference background image was subtracted under zero field conditions. Subsequent steps encompassed applying a constant in-plane field, denoted as $H_x$, up to 260 mT (for the glass substrate) and 300 mT (shapeable platform). This was accompanied by OOP pulses lasting from 0.1 s to 2 s, aiming to further expansion of the magnetic domain.

When solely subjected to an OOP field, the domain expansion resulted in a symmetrical bubble formation in all directions. When applying an in-plane field ($H_x$) an asymmetry in expansion is introduced, causing the bubble domain to take an ellipsoidal shape. This asymmetry in expansion behavior is attributed to the stabilization of Néel-type DWs over Bloch-type DWs in Pt/Co/Ta magnetic films. The application of an in-plane field disrupts the symmetry of the DWs along the x-axis, thereby influencing the propagation of the DWs either favorably or adversely in that direction. Our investigation further entailed the analysis of the DW velocity as a function of an in-plane field, both for the glass and flat polymer substrates. The Kerr microscopy images depicted in Fig. 9a, b captured the expansion of the DWs in the Pt/Co/Ta film. The grey-shaded bubble depicts the domain expansion under the -$H_z$ field. The application of an in-plane field directed to the left causes the left DW to exhibit a higher velocity compared to the right DW. Notably, when the $H_x$ field is equal to the DMI field strength (-$H_{DMI}$), the DW exhibits its minimum velocity.

The DW velocity profiles for each substrate are illustrated in Fig. 9c, d, with positive velocities indicating DW movement along the +$x$-direction. The data reveal a non-linear and monotonic trend with respect to $H_x$. For this experiment, the in-plane field strengths were constrained to 300 mT (polymer) and 260 mT (glass) to avoid multiple domain nucleation and fusions during the $H_z$ pulses. Both substrates exhibited a minimum DW velocity for field strengths surpassing 200 mT, confirming the previously reported observation[65]. However, we do not observe a clear minimum of the DW velocity within our range of

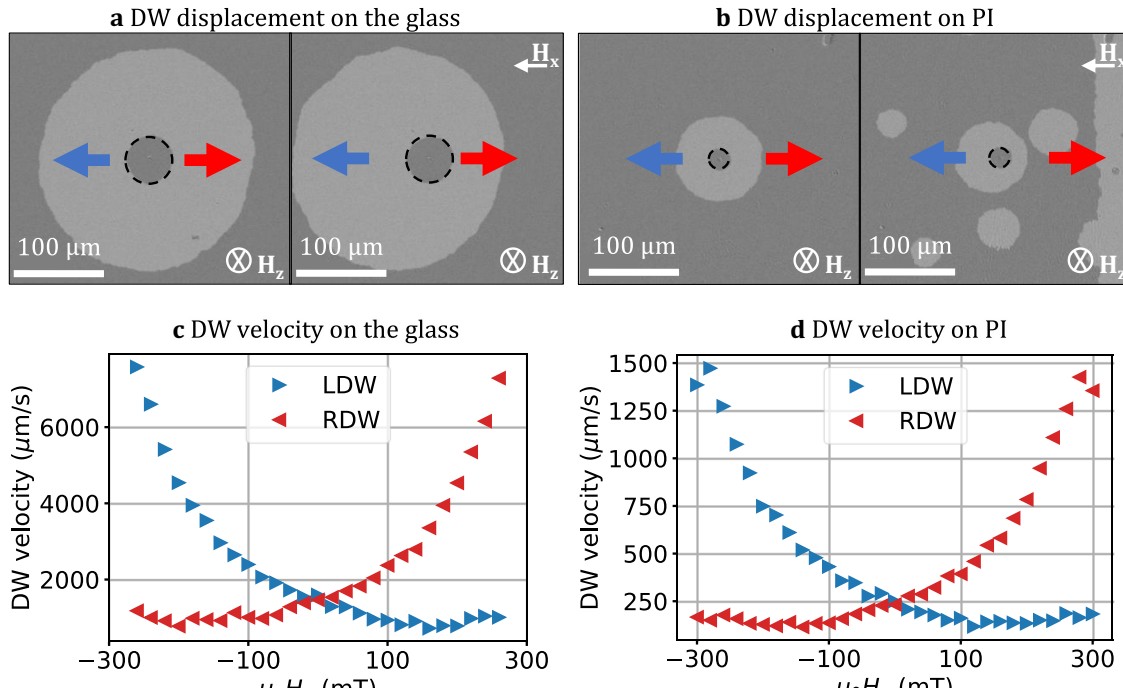

**Fig. 8 | Joule contribution analysis during current pulses. a** schematic of the measurement setup featuring a copper heater. **b** linear variation of resistance as function of temperature. **c** time-resolved alteration in temperature for a pulse width of 250 ns and current density of $1.79 \times 10^{11}$ A m$^{-2}$. **d** estimation of temperature change based on the current pulse density, considering the glass and self-assembled tube structures. The red area indicates the span of current density that was applied to move DW in 3D configuration.

**Fig. 9 | DMI measurement on glass and flat polymer.** LDW and RDW corresponds to left and right domain wall. **a, b** Bubble expansion under $H_z = 26$ mT without $H_x$ (left) and with $H_x = 100$ mT (right) field. The Field values are valid for both, figures **a** and **c**. The dashed line shows the initial position of the DW. **b, d** velocity variation for left- and right-DW (LDW and RDW) as a function of in-plane magnetic field for glass and flat polyimide substrates. The remnant $H_x$ field is smaller than 1 mT.

measurement, see Fig. 9c, d, thereby we consider $H_{DMI} > 300/260$ mT for glass/polyimide substrates. The relationship between the relative alignment of magnetic fields and the direction of bubble expansion provides evidence of a present left-handed chirality in the Neel-type DW. Previous research studies[21] demonstrated that non-trivial

curvature and induced strain significantly impact the DMI. Drawing from insights in reference[14], we anticipate that the interfacial DMI will remain relatively unchanged in the 3D state under 1.31% strain. Induced surface curvature can give rise to magnetochiral effects through the effective DMI, as elucidated in ref. 66. The influence of curvature

becomes particularly noteworthy in structures with a curvature diameter on the order of tens of nanometers. In the reported case, the tube diameter measures in the tens of micrometers. The bubble expansion method is not applicable for the 3D geometry, the Hx field will induce a parasitic nucleation of domains on the sides of the tube. To address the measurement of DMI in 3D curved geometry, one viable option is employing the Domain Wall Stray Fields technique[67,68]. This approach enables a rough measurement of the DMI parameter.

Results of field-induced DW propagation measurements show a considerable difference in the speed for the glass and flat polyimide substrates. We consider the DW speed at $H_x = 0$ mT, where the DW is driven purely with an out-of-plane field, see Fig. 9c, d. Since the stack was deposited on two substrates simultaneously, the only difference between these two samples is the surface quality, see the section Material Characterization. The DW speed on the glass is by a factor of 6 higher than on the polyimide. We attribute this behavior to the surface quality of the polyimide, which has more pinning centers in comparison to the bare glass.

## Discussion

In this study, we have demonstrated the modification of magnetic properties under induced strain and curvature that pave a new avenue for the creation of innovative three-dimensional magnetic devices. Our work demonstrates the impact of 3D geometry, strain, and curvature on current-induced DW motion and spin-orbital torque efficiency within the Pt/Co/Ta heterostructure. Utilizing a self-assembly rolling technique on a polymeric platform, we achieved field-free current-induced DW motion in a 3D racetrack configuration, reaching a maximum speed of 9 cm/s at a current density of $1.97 \times 10^{11}$ A m$^{-2}$. Our investigation delved into the influence of induced tensile strain on the spintronic framework, revealing its role in augmenting the generation of spin currents from the heavy metal layer. This augmentation leads to a noteworthy 30% enhancement in spin-orbit torque efficiency, culminating in a value of 6.9 mT 10$^{-11}$A m$^{-2}$ for the new 3D device. Simultaneously, the induced strain contributes to an increase in the pinning energy barrier within the ferromagnetic layer by enhancing the PMA anisotropy and facilitating the formation of nano-scale defects. A delicate equilibrium between these effects emerges as pivotal in determining the velocity of the domain wall in the 3D domain wall motion. It became evident that current injection significantly contributes to Joule heating, triggering a temperature rise in the racetrack that can reach $\Delta T = 76$ K. Despite this, the cumulative impact was insufficient to surmount the ascension of perpendicular magnetic anisotropy with defects formation and its associated pinning energy in the intricate 3D configuration. Furthermore, our investigation of the Dzyaloshinskii-Moriya interaction field, crucial for the stabilization of left-handed Neel walls, revealed a consistent behavior across the multilayer system on two distinct substrates. This observation underscores the minimal influence of substrate material on the strength of the DMI field.

In summary, the introduced strain and curvature-induced effects in the 3D magnetic structures transcends the current state-of-the-art paradigms. Our research results pave the way for innovative avenues in the fabrication of 3D magnetic memory devices and the exploration of strain-related phenomena in diverse magnetic materials. Notably, the entirety of our fabrication methodologies aligns seamlessly with large-scale production techniques, underscoring their profound practicality for real-world implementation.

## Methods

### Device fabrication

The nanostructures were fabricated on 1 mm thick glass substrates (D263T eco glass, SCHOTT AG Mainz, Germany). The substrates were thoroughly cleaned using a professional washer DS 500 (STEELCO S.p.A. Riese Pio, Italy) to remove any organic and inorganic

contaminants, including dust or films. The surface was then cleaned with acetone and isopropanol for 10 min each, using ultrasonic agitation, and dried with nitrogen. Subsequently, it underwent a 10-minute oxygen plasma treatment (PICO VERSION 2, IntCo, Trappes France) to eliminate any remaining residues. First, alignment crosses were prepared by the lift-off technique using AZ5214E (Microchemicals GmbH, Ulm, Germany), which was optically patterned. A Cr/Au bilayer was further sputter-deposited at room temperature and at a pressure of $1.5 \times 10^{-3}$ mbar. Next, the sample was spin-coated with a positive double-layer e-beam resist, specifically AR-P 642.04 200k and AR-P 679.03 950k. The spin-coating parameters were set at 4000 rpm/1000 rpm/60 s, and each step was followed by baking for 120 s at 180 °C. Finally, a DisCharge H2O (micro resist technology GmbH, Berlin, Germany) layer was spin-coated at 3000 rpm/1000 rpm/60 s and baked for 120 s at 90 °C. After the electron beam lithography (EBL) exposure using a dosage of 650 μC/cm², the sample was immersed in H2O for 60 s to dissolve the discharging layer, followed by development in AR 600–56 for 90 s. To create the thin film stack of Ta(3)/Pt(5)/Co(0.7)/Ta(3)/Pt(1), an ultra-high-vacuum magnetron sputtering technique was employed. The lift-off process was performed using acetone and isopropanol, with a final rinse in DI water. For the contacts and injection line, optical lift-off was carried out using the photoresist resist AZ5214E (Microchemicals GmbH, Ulm, Germany). The Ta(5)/Cu(250)/Ta(10)/Cr(5)/Au(40) contacts were deposited using magnetron sputtering equipment HZM and nanoPVD-S10A (Moorfield Nanotechnology Limited, Cheshire, UK). Finally, the resist was removed using the lift-off technique under low-power ultrasonic agitation.

### Polymeric platform

Square-shaped glass substrates measuring 50 mm×50 mm x 1 mm were used for the polymer layers. The aforementioned cleaning process was followed to ensure the glass was thoroughly cleaned. After the ultrasonic cleaning, the surface was activated using oxygen plasma in the GIGAbatch 310 M (PVA Metrology & Plasma Solutions GmbH, Wettenberg, Germany). This step further facilitated chemical surface modification with a monolayer of 3-(trimethoxysilyl)propyl methacrylate (TMSPM). To accomplish this, the glasses were placed in a vacuum oven at 150 °C for 2 h together with 250 μl of TMSPM. Subsequently, a lanthanum-acrylic acid–based organometallic photo-patternable complex was applied to the substrate. It was spin-coated at 7000 rpm, resulting in a thickness of 300 nm. The drying of the polymer was performed at 35 °C for 15 min on hotplate under the nitrogen atmosphere. and then patterned using optical lithography with the assistance of an MA6 Mask Aligner (SUSS MicroTec SE, Garching, Germany) to form the sacrificial layer (SL). To remove a water-insoluble residual and hardening the layer it was annealed at 220 °C for 20 min under the nitrogen atmosphere. A pre-heating step at 100 °C for 30 s is performed before placing the sample on a hot plate at 220 °C. Afterward, it is cooled down on a metal plate at room temperature. In a similar manner to the SL, the reinforcing polyimide (PI) layer was spin-coated on top at 4000 rpm, with prebaking temperature 50 °C, 10 min and hard baking 220 °C for 15 min, resulting in a thickness of 400 nm. It was then patterned using optical lithography. Finally, the hydrogel (HG) layer was spin-coated at 8000 rpm, with prebaking temperature 40 °C, 10 min and hard baking 220 °C for 15 min, resulting in an 800 nm thick layer. Next, it was patterned with optical lithography, to form a trapezoidal shape opening in the layer, to get the access to the PI surface.

### Self-assembly process

The sacrificial layer was selectively etched using a solution composed of DI water, 5% hydrochloric acid, and benzotriazole (all chemicals from Sigma-Aldrich Co. LLC, Germany). This etching process released the PI/HG layers from the glass substrate. In the next step, the sample was immersed in DI water for 6–12 h to initiate the swelling of the HG

layer and the formation of tubes with a larger diameter. Subsequently, the HG layer was further swollen in a solution of DI water and sodium hydroxide with a pH of 11.9, leading to the curling of the structure into a "Swiss rolls". This resulted in the formation of tight tubes with a smaller diameter. After 12 h, the tubes were gently cleaned using DI water and isopropanol (IPA) and then dried under ambient conditions. Adjusting the thickness ratio between the PI and HG layers, along with controlling the pH of the alkaline solution, allows to tune the diameter of the assembled tubes[17,69].

## Kerr microscopy

A custom-designed wide-field Kerr microscope[56] in polar configuration was employed to detect the presence of DWs. The sample was illuminated with polarized light, and the reflected light was directed through an analyzer. A LED lamp served as the light source, and the reflected light was captured by a camera (Hamamatsu, orca-spark, C11440-36U). LabView software was used to process the acquired images and obtain the final results. Prior to injecting magnetic domains or applying pulse current to move the DW, background image subtraction was performed. To prevent image drifting during the measurement process, a piezo station (Piezosystemjena NV40/3CLE, Stockholmer Str. 12 07747 Jena Germany) was utilized.

## Current-induced domain wall motion

The sample was securely attached to a standard PCB, and electrodes were bonded to 50-ohm pads. DW injection and driving were carried out using a double-channel waveform generator, Keysight 33622 A. The pulse width for DW motion was consistently set at 250 ns, and the pulse period was maintained at 1 ms to prevent cumulative Joule heating effects from impacting the DW movement. Control over the external in-plane and out-of-plane fields was achieved using the KEPCO BOP 2–20DL power supply (KEPCO, INC, SANFORD, USA). To initiate the nucleation of the DW in a magnetically saturated wire, a short voltage pulse was applied to the injection line. Subsequently, to drive the DW along the magnetic wire, a voltage pulse was applied in the desired direction.

## Temperature-dependent resistance measurements

The sample was fixed to a 1 mm thick copper plate using a silver paste, and the copper plate was then attached to the PCB using double-sided tape. An array of 50-ohm resistor heaters and a temperature sensor were mounted on top of the copper plate. To control the power output and monitor the temperature, a LakeShore 336 Cryogenic Temperature Controller (Lake Shore Cryotronics, Inc, 575 McCorkle Blvd Westerville, OH, USA) was employed. The Keysight 33622 A waveform generator was used to generate the voltage pulses. The change in stripe resistance was measured as the voltage drop across a series-connected resistor, using the Hameg HMO1024 Digital Oscilloscope.

## SOT measurements

For the SOT-efficiency measurements, a quadrupole magnetic system in conjunction with the Kerr microscope was used. In-pane and out-of-plane magnets were connected to the KEPCO BOP 2–20DL (KEPCO, INC, SANFORD, USA) and controlled by a custom-made LabVIEW script. The piezo station (Piezosystemjena NV40/3CLE, Stockholmer Str. 12 07747 Jena Germany) was utilized to avoid image drifting during the measurement process. The Keithley 2400 SMU (Tektronix, Beaverton, Oregon, United States) was used in a 4-probe regime to detect the Hall voltage during the magnetic field swiping.

## Data availability

The data that support the findings of this study are available from the corresponding author, P.F., upon request.

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

## Acknowledgements

We thank M. Bauer (IIN/ IFW Dresden) for the help with electron beam lithography. We further thank S. Nestler, and K. Leger (IIN/IFW Dresden) for their assistance with the ion milling and the polymer synthesis respectively. We appreciate the help of C. Krien and I. Fiering (IIN/IFW Dresden) for the deposition of metallic thin films. We thank C. Saggau, B. Rivkin (IIN/IFW Dresden) and Dmitriy D. Karnaushenko (MAIN, TU Chemnitz) for useful discussions. The support in the development of experimental setups from the research technology department of the Leibniz IFW Dresden and the clean room team headed by R. Engelhard (IIN/IFW Dresden) is greatly appreciated. O.G.S. acknowledges support by the German Research Foundation DFG (Gottfried Wilhelm Leibniz Prize granted in 2018, SCHM 1298/22–1). D.K. acknowledges support by the German Research Foundation DFG (KA5051/1–1 and KA 5051/3–1), as well as by the Leibniz association (Leibniz Transfer Program T62/2019).

## Author contributions

D.K. and O.G.S. conceived the idea. P.F., I.S., V.N., and D.K. designed experiments. P.F. performed experiments. P.F., I.S., V.N., and R.S. analyzed data. P.F. and D.K. wrote the manuscript with input from all authors. All authors participated in the discussions.

## Funding

## Competing interests

The authors declare no competing interests.
