## [Peer Review File · Nature Communications]

REVIEWER COMMENTS

Reviewer #1 (Remarks to the Author):

The paper demonstrates the viability of 2D and 3D domain wall racetrack using a self-assembled shapeable polymeric platform. Authors reports experimental measurements of current induced domain wall propagation, velocities, spin-orbit torque efficiencies, and temperature variations within the device. While the paper appears to have significant contribution to area of materials science, fabrication of samples, in the field of magnetism and application physics (affiliation to which is suggested by the abstract and title), it requires further research and clarification to gain a more robust understanding. The report provides a detailed description of the various stages of fabrication and experimental measurements, however, the discussion of physical effects beyond the reporting of measurement data seems unaddressed to the journal level.

My main critique lies in the paper's novelty and insufficient explanation of the underlying physical phenomena, especially about rolled 3D structures. The authors provide a general description of the observations without delving into the underlying physical phenomena in depth. Despite a meticulous presentation of their motivations and experimental methodologies, the authors fall short in discussing the results and the governing mechanisms, especially in the section "Domain wall motion in 2D planar and 3D Swiss roll-down geometries". I get the impression that the authors are conjecturing the interpretation. They claim that their work can advance racetrack technology, yet they neglect to specify the physical principles that substantiate these claims. This omission detracts from the paper's scientific rigor.

To enhance the paper's integrity and make it suitable for publication in Nature Communication, the authors should delve deeper into the physics behind their observations. They could further bolster their argument by including numerical models that corroborate their experimental results. I see no significant innovation in this manuscript besides the demonstration of the structure rolling method. The utility of rolling the structures is doubtful, particularly since the authors fail to discuss the microscopic impacts of rolling. Given these shortcomings, I have doubts about recommending the paper for publication in Nature Communication as it currently stands. However, with revisions and a more comprehensive discussion section, it may become more suitable.

Furthermore:

1. I find the literature review insufficient with regard to the 3D ferromagnetic nanostructures and novel reports about racetrack memories and domain wall motion benchmarks. A broader discussion of current reports regarding the racetrack memories, curved and/or twisted 3D ferromagnets, would significantly strengthen the readers' reference to the authors' achievements.
2. I am not really sure which part (real length) of the racetrack has been rolled up, what was the distances between the rolled layers are and whether the magnetic texture in the rolled-up part is not modified e.g. by dipole interactions. The authors also do not discuss the technical and magnetic

limitations of the rolling - what degree of rolling might physically crack the structure and what degree will modify the ferromagnetic properties?

3. More explicit presentation of system geometries should be provided at the outset (especially when structure was "rolled up"), including comparisons to the work of other research groups in the field. Furthermore, the absence of scale and dimensions in some of the figures compromises the reader's ability to accurately interpret the presented data – especially it's hard to find full dimensions of rolled geometry. This lack of detailed geometric parameters also inhibits the reproducibility of the research. Therefore, the authors should rectify these omissions to enhance the manuscript's rigor and utility. I suggest inserting a graphical sketch presenting details about geometry of measured systems in Figure 1 or 2.

4. The material parameters used in the calculations have been omitted throughout the paper. This shortcoming must be rectified.

5. Each section of the manuscript contains a fairly detailed description of sample preparation and measurement procedure. I would ask the authors to consider shortening the paper, focusing more on the physics, and thus moving the technical description to the supplementary material. In this way, a wider readership will more easily understand the content of the paper.

6. Additionally, were the roll degrees identical for each sample? If not, how does the degree of rolling affect physical properties?

Reviewer #2 (Remarks to the Author):

In this manuscript, the authors experimentally demonstrate 3D racetrack memory with a domain wall velocity of about 9 cm/s at a current density of 1.97×10^{11} A/m². Several important aspects are also investigated by the authors, including the temperature increase due to heating, the spin-orbital torque, the DMI, and the anisotropy in their 3D racetrack architecture. In my opinion, this work presents a comprehensive experimental study of 3D racetrack memory. As such, it is of great interest to the community, has technological relevance, and deserves eventual publication. However, before this manuscript is published, the authors should consider my following questions and comments/suggestions:

1, I would suggest that the authors clearly point out the distinction between their work and previous studies on 3D racetrack memory and highlight the main advancements achieved in their study in the abstract and introduction of their manuscript.

2, The authors experimentally observed a 30.1% increase in the SOT efficiency due to an applied strain of 1.3% in the rolled structure. This seems unusual to me. Can the authors provide some physical insights behind this phenomenon?

3, In the discussion of the DMI, the authors state that the DMI will remain relatively unchanged in the 3D state under 1.3% strain by drawing insights from Ref. [48]. However, there are mainly two new effects in their 3D rolled structure compared to the typical 2D geometry: one is the strain, as the author mentioned, and the other is the nontrivial curvature. To my understanding, this nontrivial curvature would lead to both effective anisotropy and DMI. For example, see the review "Magnetism in curved geometries," J. Phys. D: Appl. Phys. 49 (2016) 363001. The skyrmion textures stabilized by the curvature-induced DMI are also directly observed in the 3D magnetic system, Nature Nanotechnology 18 (3), 227-232 (2023). So, I would suggest the authors look into the DMI more carefully. Can the authors also provide a quantitative number for the change in DMI, just like the SOT efficiency?

4, There have been some very recent proposals using mobile DWs on racetracks for quantum computing, for example, in Physical Review Research 5 (3), 033166 (2023). Can the authors comment on potential ways to reduce Joule heating which can be crucial for future quantum applications? For example, perhaps using Van der Waals magnets with much lower energy scales (like anisotropies), or combining them with superconductivity and using supercurrent to drive the DW motion?

Some typos in the manuscript:

5, The first line on page3: "effects have been examined films with perpendicular and in-plane magnetic anisotropy..." -> : "effects have been examined in films with perpendicular and in-plane magnetic anisotropy..."

6, Page 5, line 10: "featuring a polymer tube with a diameter of $\varnothing 32 \mu\text{m}$..." needs to be corrected. Caption of figure 5a, and page 16 line 8 need to be fixed.

Reviewer #3 (Remarks to the Author):

The authors demonstrate a 3D rolled tube embedding a stripe shaped structures for domain wall motion characterization. These structures open the possibility of 3D racetrack memories. The impact of the fabrication requirements such as the use of polymer substrate and its surface characteristics, the rolled shape which leads to tensile strain, and the low thermal conductivities of the polymers are assessed.

Their effects on key properties such as DW velocity, DMI field, SOT efficiency and Joule heating are discussed.

Overall the results are interesting, such as the reported increase in SOT efficiency, which is in line with previous studies as ACS Nano 2021, 15, 8319. The authors team has already used similar fabrication processes for rolled structures and demonstrated its applicability to microdevices such as sensors, capacitors, inductors, etc. Altogether new opportunities may arise in spintronics devices from adding a 3D character in the design.

Still, some of the observations, data interpretation, conclusions and speculations require additional clarifications or more clear substantiations. Also, some relevant fabrication details are required to allow reproduction of the results. In the current form it does not suit publication in Nature Communications.

Comments/questions to the authors:

1) The authors indicate and show on Figure 3 that a larger H_c is obtained for the glass substrate than for polyimide coated substrates, both obtained in planar structures in continuous films and patterned stripes. What about when the polyimide is rolled? How much are H_c and PMA expected to change? Can this be observed and quantified? How will such change contribute to the final results and device performance?

A larger increase on the number of pinning centers and defects due to cracks can be expected and thus distinct magnetic behavior may be observed once rolled. Could the authors also comment on why the magnetostrictive properties of Co is not taken into account during the discussions (see also point 8).

2) Page 7-line 12, the authors mention the "... rapid thermal shrinking/expansion of polyimide during the fabrication process ..." and later "...during colling steps...". In the Materials and Methods section no curing/baking steps are mentioned for polyimide. Do these statements refer to the resist pre/post-bake for lithography? If so does this mean that the polyimide film is not completely cured?

Such thermal expansion/compression effects can also introduce cracks in the metallic thin films in both the active racetrack and the conductive electrodes, or even alter intrinsic stress contributions on the ferromagnetic thin films. Furthermore, are there any noteworthy resistivity changes in the racetrack and leads after fabrication and rolling?

The authors could elaborate on the fabrication details and possible impact on the final device performance.

3) Page 8 line 25, the authors state that the electrodes "... were made of Cr 10 nm/ Au 50 nm to prevent DW pinning." A more complete multilayer including Ta, thick Cu, Cr and Au is indicated in the Materials and Methods section. Could the authors clarify these differences and also elaborate how such a Cr/Au

structure prevents DW pinning? If related to resistivity, indicate what values are required to fulfill these specifications.

4) Page 8 line 23 the authors indicate that statistical data from 10 repetitions is obtained for the DW propagation analysis. These should be shared with the reader. This would allow to evaluate dispersion on the measured properties and the real impact of non-uniformities which may arise from materials growth or fabrication steps.

Furthermore, examples of all measurements for more than one device, with similar geometrical parameters, should be shared, allowing evaluation of reproducibility and consistency of the results obtained. The same is applicable to the results in Figure 6 (see point 11).

5) Page 9 line 14: Could the authors clarify what is meant by a "... 3/4 of the turn has been achieved after the tube rolling process" and why this is relevant for the measurements and analysis being performed.

6) In page 10 and Figure 5, the authors mentioned that a fitting to the creep law is done to the data of fig 5d. Could the authors elaborate on what, if any, quantitative information is extracted from this analysis, or if relevant conclusions can be drawn, and what fitting parameters are considered?

7) Page 10 Lines 24-25 the authors state that the observed "... reduction in the DW velocity in the 3D racetrack device .. is rather counterintuitive." Could the authors elaborate why so?

Overall, if more intrinsic defects are observed for 3D rolled structures, namely the appearance of cracks, there should be an increase in the number of pinning center for DWs. Furthermore, the effects of all fabrication steps on the properties of the thin films microstructures (magnetic, electric and morphological) could also be expected to hinder DW movement as it may accumulate different type of defects and smear the effective anisotropies.

In fact, in page 20 line 26 the authors state that the "... DW speed on the glass is by a factor of 6 higher than on polyimide" which they attribute to the surface quality and more pinning centers compared to glass.

The appearance of micro/nano cracks in bendable metallic stripes on polymers, their impact in $M(H)$ and resistivity are well reported. It seems that the reduction in DW velocity once the racetrack is built on the polymeric platform could be anticipated? Also, could the authors elaborate and link these results to the H_c values observed in Fig.1d, and the potential impact of increasing resistivity of the racetrack and electrical leads due to cracking on the DW movement? Are any of these defects visible under the optical/SEM microscope?

Finally, on page 18 line 23-24, two suggestions are given to enhance the DW speed in the rolled tube. Could the authors comment on why encapsulation (for neutral axis shifting) commonly used to minimize cracks or strain in flexible structures is not considered? Is there a disadvantage for these devices?

8) Page 11 line 2: the authors mention a significant H_c increase in the magnetic stripe upon rolling, but no results are shown to support this statement. Furthermore, the authors also assume an increase in PMA. A more convincing way should be used to support these claims. Also, could the authors comment on the role of magnetostriction of Co in these structures? Depending on the degree of bending and on the direction of applied stress, competing anisotropies (e.g. magnetoelastic, surface, dipolar) can lead to a change in the easy axis orientation. If such effects can be neglected in these stripes, the authors should comment why.

9) Page 19 line 12: Figures 1f and 1e are mentioned in the text but not available in the manuscript. Furthermore, additional schemes clarifying what is meant by tensile strain, rolling down, the positioning of the racetrack for Kerr microscopy analyses, could be added to facilitate visualization.

10) On Page 15 line 18 the authors say “The accuracy of this alignment is confirmed through hysteresis loop measurements. The Hall voltage is the same for an out of plane field of 0 Oe and 200 Oe ...”. This is an important step to ensure that the measurements are performed in comparable conditions and that no undesired components of the magnetic field are acting on the device. The authors are strongly encouraged to share these data/results with the readers.

11) For figures 6b and 6d, the discussion indicates that the hysteresis loops of the rolled structure “... exhibit a similar appearance compared to the planar case”. However, both the shape of the curves, width of the loop and its polarization are all strikingly different. This seem to indicate that quite different magnetic/electrical properties are present in each structure. Please comment or clarify this reasoning.

Again, results for more than one device result should be presented.

12) Page 16 line 7: the authors give the estimation of SOT efficiency but no framing within current most up-to-date results in the field are indicated especially for the planar structures which can be directly comparable.

13) Page 18 line 16: the authors indicate that the temperature rise “does not boost the DW energy enough to overcome the additional pinning energy (barriers) arising from the strain induced PMA modification.” Have the authors considered other changes due to temperature rise that may also justify this behavior? For example, the use of highly temperature sensitive polymer substrates which may have implications in different strains as a function of temperature, or the change of effective anisotropy as a function of temperature.

Could the authors comment if this or similar effects may be relevant for this study and envisaged final application, or if not why.

14) The authors then proceed to characterize the DMI field via bubble expansion technique imaged by Kerr microscopy. The authors should clearly identify in Figure 9 and in the text if such studies are done in a planar geometry or rolled structure for the polymer substrate. Although studies in full planar structures grow on polyimide can help address some of the underlying roots of the decrease in the DW velocity and support the assessment of DMI field, the real impact of all variables and the most relevant measurement would come from 3D rolled structures being in the same conditions as the final device. In the planar polymer there is no strain from bending (only intrinsic) and no induced cracks.

In page 20 line 20 the authors then state that they “anticipate that the interfacial DMI will remain relatively unchanged in the 3D state under 1.3% strain.” This statement is not clearly substantiated. Once under strain it is not clear how several factors such as surface anisotropy, additional defects, surface changes, etc, are expected to stay the same. The authors should elaborate, clarify, or support further with literature their statement. Ref 48 is a start but the differences in the fabrications methods, eg substrate surface characteristics, critical interfaces with Pt and Ir, or the size of the measured structures are quite different.

15) The use of CGS and SI units through the manuscript should be revised. The authors indicate several times H in Oe and $\mu_0 H$ in Oe. Especially in the graphics there is no consistency in labeling the magnetic field (e.g. figure 6F and figure 6D). Systematic use of SI units for magnetic field should be implemented.

16) Materials and Methods

- Page 27 line 17: the authors mentioned the spin-coating of a discharging layer for the EBL process. The materials, or types of polymers used should be indicated.
- Page 28 lines 12-17: No mentioning of baking steps are given for the polymers in particular regarding polyimide. It is quite well known that different curing temperatures can be used, but depends significantly on the type of polyimide that is being employed. This information should be included.

Also, the hydrogel is said to be spin-coated with a trapezoidal shape. Maybe such shape is obtained by subsequent lithography. If so the authors should elaborate on which steps of mask definition and pattern transfer conditions are used.

These are important information for the work to be reproduced.

17) Minor typos:

- page 3 line 1: “has been examined in films”
- page 3 line 21: “ velocities at the angular momentum compensation temperature”
- page 13 line 13: “as a function of I_{DC} ”

- page 27 line 14: units of pressure are missing
- page 20 line 17" units of fields are missing in H_DMI

We express our sincere gratitude for the time and effort invested by the reviewers in evaluating our manuscript. In the subsequent sections, we reference the reviewers' comments (in black). We are confident that we have addressed all the raised questions, which are indicated in blue in both this response letter and the manuscript text.

Reviewer #1 (Remarks to the Author):

The paper demonstrates the viability of 2D and 3D domain wall racetrack using a self-assembled shapeable polymeric platform. Authors reports experimental measurements of current induced domain wall propagation, velocities, spin-orbit torque efficiencies, and temperature variations within the device. While the paper appears to have significant contribution to area of materials science, fabrication of samples, in the field of magnetism and application physics (affiliation to which is suggested by the abstract and title), it requires further research and clarification to gain a more robust understanding. The report provides a detailed description of the various stages of fabrication and experimental measurements, however, the discussion of physical effects beyond the reporting of measurement data seems unaddressed to the journal level.

My main critique lies in the paper's novelty and insufficient explanation of the underlying physical phenomena, especially about rolled 3D structures.

The authors provide a general description of the observations without delving into the underlying physical phenomena in depth. Despite a meticulous presentation of their motivations and experimental methodologies, the authors fall short in discussing the results and the governing mechanisms, especially in the section "Domain wall motion in 2D planar and 3D Swiss roll-down geometries". I get the impression that the authors are conjecturing the interpretation.

They claim that their work can advance racetrack technology, yet they neglect to specify the physical principles that substantiate these claims. This omission detracts from the paper's scientific rigor. To enhance the paper's integrity and make it suitable for publication in Nature Communication, the authors should delve deeper into the physics behind their observations. They could further bolster their argument by including numerical models that corroborate their experimental results.

I see no significant innovation in this manuscript besides the demonstration of the structure rolling method. The utility of rolling the structures is doubtful, particularly since the authors fail to discuss the microscopic impacts of rolling.

Given these shortcomings, I have doubts about recommending the paper for publication in Nature Communication as it currently stands. However, with revisions and a more comprehensive discussion section, it may become more suitable.

Response: We express our gratitude to the reviewer for the constructive feedback on the clarity of our work. The revised abstract, title, and introduction now distinctly highlight the contribution of our work to the field of materials science, particularly in the context of building 3D memory units based on racetrack architecture. We acknowledge the importance of clarifying the primary objective of our study, which is to showcase a proof-of-concept for 3D thin-film racetrack devices, rather than an exhaustive exploration of strain-induced effects in magnetic heterostructures. In our revised version, we have aimed to mitigate any potential misinterpretation by emphasizing the complexity associated with implementing 3D thin-film racetrack devices. Our focus extends to the successful demonstration of 3D current-induced domain wall motion through a self-assembly technique, integrating critical functional elements such as information writing and reading. To the best of our knowledge, a complete memory unit (write, store, read) in the field of 3D DW racetrack memory has not been reported, adding a significant dimension to the advancement of racetrack technology.

Concerning the physical description of the observed results: Our analysis delves into critical facets of the self-assembly process and elucidates the influence of 3D geometry on the performance of magnetic layer systems, drawing insights from previous studies. We described all physical effects on the accepted level to be available for understanding for majority of readers. Also, the particular description of the mentioned and measured effects in the manuscript (SOT, DMI, domain wall motion and its speed reduction) can be found in the introduction part of the revised text, page 3 line 7-9. These effects were already discussed in the number of previous research papers that are cited, page 3 line 9. We agree with the reviewer that numerical model is favored for this project. Thereby, we have performed a micromagnetic simulation with MuMax3 code, to support our claim, that the increase of PMA will lead to the DW speed reduction. We assumed an exchange constant of $A = 20\text{pJ/m}$, a uniaxial anisotropy of $K_u = 1.38\text{ MJ/m}^3$, and saturation magnetization $M_s = 1.19\text{ MA/m}$, these parameters were taken from the recent study⁶³. For the DMI we took a value of 0.75 mJ/m^2 and we assumed gilbert damping constant $\alpha = 0.48$. Disorders were simulated with a random grain structures with local anisotropy variation and reduction of the exchange coupling between neighboring grains. Grain structure were simulated with Voronoi tessellation with average size of 8 nm [APPLIED PHYSICS LETTERS 110, 132404 (2017)]. We model a system with dimensions of $64 \times 64 \times 1\text{ nm}$ with z cell size of 1 nm . Periodic boundary conditions are assumed along z and y [J. Appl. Phys. 134, 171101 (2023); doi: 10.1063/5.0160988]. SOT was implemented into the model, with current values, as in the experimental part.

Domain wall speed as function of current for the flat glass and rolled tube. Uniaxial anisotropy for 2D configuration is $K_u = 1.38 \text{ MJ/m}^3$, and for 3D is $1.5 \times 1.38 \text{ MJ/m}^3$.

A nanostrip model, $64 \times 64 \text{ nm}$ with local magnetization vectors, during the applied current. Blue color – Neel domain wall between $\pm z$ states.

Results of the simulation indicate a reduction in DW speed of approximately two-fold when the K_u constant is increased by 50%. This finding corroborates the assertion made in the main body of the report. While the reduction in DW speed does not correspond exactly to the experimental results, the underlying trend remains consistent. Further modeling is necessary to incorporate the intricate energy landscape

behavior that arises during the rolling process. However, to cover all aspects of this work, an additional work has to be done, that goes beyond one manuscript and can be published as independent research paper. As we show the proof-of-concept, we present a general and principal explanation of the observed effects. Finally, expanding 3D self-assembly technologies into the magnetic data storage sector represents a logical progression, given the significant limitations of the 2D configuration in memory devices. This discussion was added as a support material, to the main text, page 12, line 1.

Furthermore:

I find the literature review insufficient with regard to the 3D ferromagnetic nanostructures and novel reports about racetrack memories and domain wall motion benchmarks. A broader discussion of current reports regarding the racetrack memories, curved and/or twisted 3D ferromagnets, would significantly strengthen the readers' reference to the authors' achievements.

Response: We appreciate the reviewer's observation. Recognizing the validity of this point, we have implemented a corresponding correction in the text, located on page 3, line 3.

I am not really sure which part (real length) of the racetrack has been rolled up, what was the distances between the rolled layers are and whether the magnetic texture in the rolled-up part is not modified e.g. by dipole interactions. The authors also do not discuss the technical and magnetic limitations of the rolling - what degree of rolling might physically crack the structure and what degree will modify the ferromagnetic properties?

Response: We thank the reviewer for the comment. The distance between the rolled layers measures 900 nm (600/300 – PI/HG). The ferromagnetic (FM) stripe was intentionally not completely rolled into the ring, preventing dipole interactions between different parts of the stripe. We anticipate that our structure can endure a maximum strain of up to 4.5% [*Nano Lett.* 2011, 11, 6, 2522–2526] before experiencing fractures. Beyond this threshold, electrical contact will cease. The ferromagnetic properties of the film/stripe atop the polymer will promptly change under applied strain, with the extent of this alteration restricted by the detectivity of the observation tool. The observable shift in FM properties will occur once the signal surpasses the detectivity threshold. Nevertheless, we are planning to make systematic measurements of the resistivity behaviour under the applied strain.

More explicit presentation of system geometries should be provided at the outset (especially when structure was "rolled up"), including comparisons to the work of other research groups in the field. Furthermore, the absence of scale and dimensions in some of the figures compromises the reader's ability

to accurately interpret the presented data – especially it's hard to find full dimensions of rolled geometry. This lack of detailed geometric parameters also inhibits the reproducibility of the research. Therefore, the authors should rectify these omissions to enhance the manuscript's rigor and utility. I suggest inserting a graphical sketch presenting details about geometry of measured systems in Figure 1 or 2.

Response: We thank reviewer for this comment. The appropriate photo was added into the manuscript, Figure 2 a.

The material parameters used in the calculations have been omitted throughout the paper. This shortcoming must be rectified.

Response: We thank the reviewer for this comment. The corresponding calculation for the induced strain was added to the supplementary part, as supplementary note 1.

Each section of the manuscript contains a fairly detailed description of sample preparation and measurement procedure. I would ask the authors to consider shortening the paper, focusing more on the physics, and thus moving the technical description to the supplementary material. In this way, a wider readership will more easily understand the content of the paper.

Response: We thank the reviewer for this comment. We have elaborated the text accordingly. The technical description from **Material characterization and self-assembled microtubular structures (now it is called Material characterization)** was moved to the Materials and Methods part.

Additionally, were the roll degrees identical for each sample? If not, how does the degree of rolling affect physical properties?

Response: We express our gratitude to the reviewer for the comment. The degree of rolling was specifically 5 turns for the tube with a diameter of 32 μm . The measurement of the domain wall (DW) speed was conducted on a tube with a 32 μm diameter, while the spin-orbit torque (SOT) measurements were performed on a tube with a slightly smaller diameter of 30.9 μm . We chose to omit this minor tube variation in our paper, as it is not anticipated to have a significant impact on the final results. Small variation of tube diameter ~ 0.035 will induce a change in the strain in 4th decimal place.

Conclusively, we appreciate the constructive remarks provided by the reviewer, which have contributed to our efforts in enhancing the clarity and quality of the text.

Reviewer #2 (Remarks to the Author):

In this manuscript, the authors experimentally demonstrate 3D racetrack memory with a domain wall velocity of about 9 cm/s at a current density of 1.97×10^{11} A/m². Several important aspects are also investigated by the authors, including the temperature increase due to heating, the spin-orbital torque, the DMI, and the anisotropy in their 3D racetrack architecture. In my opinion, this work presents a comprehensive experimental study of 3D racetrack memory. As such, it is of great interest to the community, has technological relevance, and deserves eventual publication. However, before this manuscript is published, the authors should consider my following questions and comments/suggestions:

Response: We thank the reviewer for the positive remark.

I would suggest that the authors clearly point out the distinction between their work and previous studies on 3D racetrack memory and highlight the main advancements achieved in their study in the abstract and introduction of their manuscript.

Response: We express our appreciation to the reviewer for this valuable comment. We have incorporated the necessary changes as suggested. Specifically, modifications have been made in the Abstract, lines 22-23 and 24-25, as well as in the introduction section on page 3, lines 3. For your convenience, we are also replicating the text here:

1. So far, most of the studies with 3D magnetic structures were performed in the helices and nanowires, mainly with stationary DWs. Notably, we present a complete 3D memory unit with write, read and store functionality.
2. Recent reports have delved into several avenues to explore the DW motion in 3D geometries^{7,11,14,70}. The propagation of DWs in curved nanowire was theoretically studied in the cylindrical magnetic nanowires, revealing non-trivial oscillatory behavior, that depends on the curvature of the wire^{8,9,10}. Several research groups presented studies on DWs and magnetic textures in 3D ferromagnetic nanohelices^{11,12,13} fabricated with focused electron beam induced deposition. Recent report¹⁴ further showcased the realization of current-induced domain wall motion in 3D racetrack (RT) based on synthetic antiferromagnets with PMA, fabricated on a free-standing tilted membrane. These domains can be efficiently manipulated through the application of ultra-short current pulses^{15,16}. The RT stands out for its immense potential to achieve significantly higher data density storage compared to other emerging memory technologies. A crucial aspect of the RT is its ability to reliably manipulate data at high speed while consuming low power^{17,18}. This unique combination of features positions the RT as a leading candidate for the

realization of next-generation memory devices, offering exceptional performance and non-volatility.

The authors experimentally observed a 30.1% increase in the SOT efficiency due to an applied strain of 1.3% in the rolled structure. This seems unusual to me. Can the authors provide some physical insights behind this phenomenon?

Response: We appreciate the reviewer's comment, and in response, we have included the relevant text on page 17, lines 9-17. For your convenience, we are also replicating the text here:

As the applied strain is increased, the racetrack stripe gets narrower, consequently resulting in an increase in the metal's resistivity. According to the discussion⁵¹, the heightened efficiency of SOT efficiency under elongation strain is attributed to extrinsic scattering. It was reported that in heavy metals, like Pt, Ta, W, Pd the intrinsic scattering typically dominates over the extrinsic. The total spin Hall conductivity can be expressed as a sum of the intrinsic and extrinsic parts. The total spin Hall conductivity, exhibits the increase in response to the applied elongation strain. In the systematic analysis, it was shown that the intrinsic part remains constant as function of strain. This result led to the conclusion that SOT modulation is attributed to the extrinsic part.

In the discussion of the DMI, the authors state that the DMI will remain relatively unchanged in the 3D state under 1.3% strain by drawing insights from Ref. [48]. However, there are mainly two new effects in their 3D rolled structure compared to the typical 2D geometry: one is the strain, as the author mentioned, and the other is the nontrivial curvature.

Response: We appreciate the reviewer's comment. It is acknowledged that the DMI (Dzyaloshinskii-Moriya interaction) undergoes variations in nontrivial geometries, as discussed in the reference [Robert Streubel et al., 2016, *J. Phys. D: Appl. Phys.* 49 363001, "Magnetism in curved geometries"]. The impact of curvature becomes significant in structures with a curvature diameter on the order of tens of nanometers. In our specific case, the tube diameter is 16 μm , leading us to infer that the DMI does not undergo a substantial variation in our experimental setup. We have extended discussion accordingly in the main text. Page 22, lines 15-17 and 18-20.

To my understanding, this nontrivial curvature would lead to both effective anisotropy and DMI. For example, see the review "Magnetism in curved geometries," *J. Phys. D: Appl. Phys.* 49 (2016) 363001. The skyrmion textures stabilized by the curvature-induced DMI are also directly observed in the 3D magnetic

system, *Nature Nanotechnology* 18 (3), 227-232 (2023). So, I would suggest the authors look into the DMI more carefully. Can the authors also provide a quantitative number for the change in DMI, just like the SOT efficiency?

Response: We appreciate the reviewer's emphasis on the significance of quantitative values for the DMI. Unfortunately, obtaining these specific numbers proves challenging due to the nature of the bubble expansion method. This method precludes the measurement of DMI changes (perpendicular to the rolling direction of the tube) in a rolled geometry. In the case of thin films, parasitic nucleation on the sides of the tube, perpendicular to the applied in-plane field, can occur. While we contemplate patterning the film to create a circular structure on the tube's dome, free from shape changes, we encounter challenges. The circuit size becomes a limiting factor, making it too small for reliable observation of magnetic domain dynamics. Additionally, utilizing ultra-short magnetic field pulses would necessitate the development and integration of an additional micro-coil, which poses a consideration for future studies. To address the measurement of DMI in 3D curved geometry, one viable option is employing the Domain Wall Stray Fields technique [arXiv:2009.11830v3], [*PHYSICAL REVIEW B* 94, 064413 (2016)]. This approach enables a rough measurement of the DMI parameter. We have incorporated this discussion into the text to enhance clarity on the challenges and potential avenues for further exploration, page 22 line 18. For your convenience, we are also replicating the text here:

Previous research studies [J. Phys. D: Appl. Phys. 49 363001, "Magnetism in curved geometries"] demonstrated that non-trivial curvature and induced strain significantly impact the DMI. Drawing from insights in reference⁴⁸, we anticipate that the interfacial DMI will remain relatively unchanged in the 3D state under 1.31% strain. Induced surface curvature can give rise to magnetochiral effects through the effective DMI, as elucidated in reference [Nature Nanotechnology volume 18, 227–232 (2023)]. The influence of curvature becomes particularly noteworthy in structures with a curvature diameter on the order of tens of nanometers. In the reported case, the tube diameter measures in the tens of micrometers. Bubble expansion method is not applicable for the 3D geometry, the H_x field will induce a parasitic nucleation of domains on the sides of the tube. To address the measurement of DMI in 3D curved geometry, one viable option is employing the Domain Wall Stray Fields technique [arXiv:2009.11830v3], [*PHYSICAL REVIEW B* 94, 064413 (2016)]. This approach enables a rough measurement of the DMI parameter.

There have been some very recent proposals using mobile DWs on racetracks for quantum computing, for example, in *Physical Review Research* 5 (3), 033166 (2023). Can the authors comment on potential ways to reduce Joule heating which can be crucial for future quantum applications? For example, perhaps using

Van der Waals magnets with much lower energy scales (like anisotropies), or combining them with superconductivity and using supercurrent to drive the DW motion?

Response: We greatly appreciate this insightful comment from the reviewer. However, we must acknowledge that this question lies beyond the scope of our expertise. In the paper referenced, the measurements are intended to be conducted at temperatures ranging from 50-100 mK, which significantly deviates from the temperature variations discussed in our work. Furthermore, the approaches and technical solutions for thermal insulation differ substantially for typical room temperature measurements.

While we acknowledge the potential interest in such inquiries within the scientific community, we have included a brief remark in the introduction section, highlighting the reference and an intriguing quantum entanglement application scenario for racetrack devices.

There are several ways to reduce the Joule contribution, which can be categorized into two groups: the utilization of new materials (material science) and the engineering optimization of the system.

In the first group, solutions include the use of metals with low resistivity, while maintaining high SHA [*Phys Rev Mat* 6, 074206, 2022] as a source of spin-polarized current. Additionally, it is crucial to employ spin-triplet supercurrent with a non-zero net spin [*Chin Phys Lett* Vol 35, 7, 077401, 2018]. Finally, a similar approach to the superconductive resonator can be employed to address the Joule heating issue [Alex Gurevich 2023 *Supercond. Sci. Technol.* 36 063002].

On the other hand, it is possible to develop an efficient cooling approach. For example, utilizing active helium cooling alongside a contact cooling approach. The most effective strategy ultimately depends on the operational temperature.

Some typos in the manuscript:

The first line on page3: “effects have been examined films with perpendicular and in-plane magnetic anisotropy...” -> : “effects have been examined in films with perpendicular and in-plane magnetic anisotropy...”. Page 5, line 10: “featuring a polymer tube with a diameter of $\text{\O}32 \mu\text{m}$...” needs to be corrected. Caption of figure 5a, and page 16 line 8 need to be fixed.

Response: We express our gratitude to the reviewer for identifying this typo. It has been promptly corrected as per your observation.

Reviewer #3 (Remarks to the Author):

The authors demonstrate a 3D rolled tube embedding a stripe shaped structures for domain wall motion characterization. These structures open the possibility of 3D racetrack memories. The impact of the fabrication requirements such as the use of polymer substrate and its surface characteristics, the rolled shape which leads to tensile strain, and the low thermal conductivities of the polymers are assessed. Their effects on key properties such as DW velocity, DMI field, SOT efficiency and Joule heating are discussed. Overall the results are interesting, such as the reported increase in SOT efficiency, which is in line with previous studies as ACS Nano 2021, 15, 8319. The author's team has already used similar fabrication processes for rolled structures and demonstrated its applicability to microdevices such as sensors, capacitors, inductors, etc. Altogether new opportunities may arise in spintronics devices from adding a 3D character in the design.

Response: We thank the reviewer for the positive remark.

Still, some of the observations, data interpretation, conclusions and speculations require additional clarifications or more clear substantiations. Also, some relevant fabrication details are required to allow reproduction of the results. In the current form it does not suit publication in Nature Communications.

Comments/questions to the authors:

The authors indicate and show on Figure 3 that a larger H_c is obtained for the glass substrate than for polyimide coated substrates, both obtained in planar structures in continuous films and patterned stripes. What about when the polyimide is rolled? How much are H_c and PMA expected to change? Can this be observed and quantified? How will such change contribute to the final results and device performance?

Response: We appreciate the reviewer's inquiries. Following the rolling of polyimide, the coercivity (H_c) undergoes an average increase of 17.5%, a change measured using MOKE. Estimating the variation in perpendicular magnetic anisotropy (PMA) poses challenges. To assess PMA, one typically measures an anisotropy field. However, two obstacles arise in this context. Firstly, the in-plane field required exceeds the technical capabilities of our instruments. Secondly, the 3D geometry introduces the complication of domain wall nucleation on the side of the tubular/curved ferromagnetic thin film. The increase in coercivity (H_c) correlates with a reduction in domain wall speed^{48,49}, reference from the main text. We have added this remark to the main text, page 12 line 15.

A larger increase on the number of pinning centers and defects due to cracks can be expected and thus

distinct magnetic behavior may be observed once rolled. Could the authors also comment on why the magnetostrictive properties of Co is not taken into account during the discussions (see also point 8).

Response: We appreciate the reviewer for bringing this omission to our attention. Typically, the magnetostrictions of ferromagnetic materials are addressed and discussed in sections dedicated to estimating the effective anisotropy (K_{eff}) change. Thanks to your insightful comment, we have rectified this oversight on page 12, lines 15-18.

Page 7-line 12, the authors mention the "... rapid thermal shrinking/expansion of polyimide during the fabrication process ..." and later "...during colling steps...". In the Materials and Methods section no curing/baking steps are mentioned for polyimide. Do these statements refer to the resist pre/post-bake for lithography? If so does this mean that the polyimide film is not completely cured?

Response: We appreciate the reviewer's thorough question. The mentioned section pertains to the fabrication of the polymer platform. After spin-coating the polymers, a pre-baking process is employed at a low temperature, below 100°C, to eliminate dimethylacetamide, the primary solvent for polyamic acid. Subsequently, lithography is performed, during which the polyamic acid is crosslinked at grafted ionic groups. The development stage then removes non-crosslinked polyamic acid. The final step involves a hard baking process at 220°C for 20 minutes to ensure complete imidization of the polyamic acid, during which the polymer undergoes chemical transformation, dehydration, and consequently, shrinkage. A pre-heating step at 100°C for 30 seconds is followed by placing the sample on a hot plate at 220°C. Afterward, it is cooled down on a metal plate at room temperature. This clarification has been incorporated into the Material section of the paper and is highlighted for emphasis.

Such thermal expansion/compression effects can also introduce cracks in the metallic thin films in both the active racetrack and the conductive electrodes, or even alter intrinsic stress contributions on the ferromagnetic thin films. Furthermore, are there any noteworthy resistivity changes in the racetrack and leads after fabrication and rolling?

Response: We acknowledge the reviewer's concern. Nevertheless, the fabrication of the metal stacks is conducted on a polyimide surface that has already undergone imidization and relaxation. Throughout the ferromagnetic layer patterning steps, the sample is not heated beyond 120°C, and we assert that this temperature treatment does not significantly impact the resistivity of the ferromagnetic material and the associated metal leads. After the rolling process we observed that change of the resistance by 0.56%, based on measurements of 3 devices.

The authors could elaborate on the fabrication details and possible impact on the final device performance.

Response: We appreciate the reviewer for bringing up this concern. It is worth noting that all fabrication steps for device patterning are standard and widely documented in the literature, with a minimal impact on the final device's performance. The oversight related to the heat treatment of the polymer platform has been addressed, and corresponding text has been added to the Materials and Methods section.

Page 8 line 25, the authors state that the electrodes "... were made of Cr 10 nm/ Au 50 nm to prevent DW pinning." A more complete multilayer including Ta, thick Cu, Cr and Au is indicated in the Materials and Methods section. Could the authors clarify these differences and also elaborate how such a Cr/Au structure prevents DW pinning? If related to resistivity, indicate what values are required to fulfill these specifications.

Response: We appreciate the reviewer's question. Regarding pinning, the comparison was made with magnetic contacts. Some research groups utilize magnetic contacts, etched from the same stack as the racetrack, to measure the change in Hall voltage resulting from the arrival of the DW. Our design incorporates several functional electrode structures: those for DW detection and for DW nucleation. Cu, Cr, and Ta were chosen for the injection line to enable the injection of high current for locally creating magnetic domains. To prevent de-attachment from the surface during the rolling process, the electrodes for DW detection were intentionally made thin. We have added clarifying text to elaborate on this aspect, page 5 line 12 and page 9 line 13.

Page 8 line 23 the authors indicate that statistical data from 10 repetitions is obtained for the DW propagation analysis. These should be shared with the reader. This would allow to evaluate dispersion on the measured properties and the real impact of non-uniformities which may arise from materials growth or fabrication steps.

Furthermore, examples of all measurements for more than one device, with similar geometrical parameters, should be shared, allowing evaluation of reproducibility and consistency of the results obtained. The same is applicable to the results in Figure 6 (see point 11).

Response: We appreciate the reviewer for bringing attention to this. We have updated the data availability statement, which now reads as: "The data that support the findings of this study are available from the corresponding author, P.F., upon reasonable request."

Page 9 line 14: Could the authors clarify what is meant by a “... 3/4 of the turn has been achieved after the tube rolling process” and why this is relevant for the measurements and analysis being performed.

Response: We appreciate the reviewer for noting this crucial detail. In our experiment, we encountered challenges in achieving the intended tube diameter of 20 μm , resulting in a diameter of only 32 μm . Consequently, the racetrack, designed with a length of 70 μm , did not form a complete ring but rather an arc. In this configuration, there is no interaction between overlapping layers of the racetrack, simplifying the analysis of domain wall (DW) behavior. We have made slight adjustments to the text to prevent potential misunderstandings.

In page 10 and Figure 5, the authors mentioned that a fitting to the creep law is done to the data of fig 5d. Could the authors elaborate on what, if any, quantitative information is extracted from this analysis, or if relevant conclusions can be drawn, and what fitting parameters are considered?

Response: the fitting parameters that were considered: $\zeta = U_c/kT (1/H_{dep})^\mu$ and V_0 . These parameters are functions of the pinning properties of the sample, determined by the correlation length and pinning strength of the disorder [Phys. Rev. Lett. 80, 849–852, 1998]. Below is the table containing the fitting parameters obtained from our measurements:

	2D glass	3D tube
V_0	21030.9	44529.9
ζ	32.803	64.390

However, these parameters need to be approached with caution. Firstly, the temperature in the fitting parameter is not constant, unlike when fitting is performed for field-driven domain wall (DW) motion. Secondly, the substrate properties differ. We believe that a more detailed follow-up investigation is necessary to draw reasonable conclusions from these numbers.

Page 10 Lines 24-25 the authors state that the observed “... reduction in the DW velocity in the 3D racetrack device .. is rather counterintuitive.” Could the authors elaborate why so?

Response: We are grateful for the meticulous attention to important details. Initially, we anticipated that the thermal contribution (where the stripe heats more on the polymer than on the glass) would aid in DW propagation, resulting in a higher speed in the rolled state. However, our findings indicate that the change in PMA is more pronounced than the additional contribution from Joule heating. We have further emphasized this aspect in the text, page 12 line 12.

Overall, if more intrinsic defects are observed for 3D rolled structures, namely the appearance of cracks, there should be an increase in the number of pinning center for DWs. Furthermore, the effects of all fabrication steps on the properties of the thin films microstructures (magnetic, electric and morphological) could also be expected to hinder DW movement as it may accumulate different type of defects and smear the effective anisotropies.

Response: We express gratitude to the reviewer once again. The fabrication process for the racetrack on both glass and polymer platforms is identical. Consequently, the contribution of fabrication steps remains consistent.

In fact, in page 20 line 26 the authors state that the “... DW speed on the glass is by a factor of 6 higher than on polyimide” which they attribute to the surface quality and more pinning centers compared to glass.

Response: Thank you for bringing up this comment. In this section, we specifically address the field-driven DW motion, where there is no Joule contribution. We emphasize this point in the text.

The appearance of micro/nano cracks in bendable metallic stripes on polymers, their impact in $M(H)$ and resistivity are well reported. It seems that the reduction in DW velocity once the racetrack is built on the polymeric platform could be anticipated?

Response: Thank you for bringing up this comment. Apparently, our measurements show that the current induced DW speed of the flat polymer is higher, then on the glass, despite the fact that flat polymer is unperfect and creates numerous pinning centers. Alongside with DW speed measurements, we estimated the temperature variation of the RT on different substrates. On the following graphs these dependences are shown.

Figure: CIDWS for the glass and flat polymer substrates. Here, the maximum current was limited by the random nucleation of magnetic domains in the stripe.

Figure: RT's temperature change for the glass and flat polymer substrates. The temperature variation was measured for wider range of the current density.

We believe that temperature plays a key contribution in DWS enhancement, its contribution is strong enough to overcome the additional pinning that is coming from the polymer substrate. (These results were obtained on the 500 nm wide racetrack).

Also, could the authors elaborate and link these results to the H_c values observed in Fig.1d, and the potential impact of increasing resistivity of the racetrack and electrical leads due to cracking on the DW movement? Are any of these defects visible under the optical/SEM microscope?

Response: We appreciate the reviewer's concern. These defects are not visible in either optical or SEM microscopy. We consider that H_c is governed mainly by the of pinning/nucleation mechanism in the etched structures, in the reported case. On the flat polymer we do not observe considerable and systematic increase of the resistivity. And we do not expect or saw any cracks on the metal deposited on the flat polymer. Here, we present our old AFM scan, when we were optimizing our fabrication process. On the figure, you see the section of the FM stripe, deposited on the PI layer. Even, the scanning area can be considered too large and its only one frame, still we do not observe cracking on the flat polymer.

On the rolled polymer, the can not observe the cracking, due to charging of the polymer tube.

Finally, on page 18 line 23-24, two suggestions are given to enhance the DW speed in the rolled tube.

Could the authors comment on why encapsulation (for neutral axis shifting) commonly used to minimize cracks or strain in flexible structures is not considered? Is there a disadvantage for these devices?

Response: Certainly, employing an additional layer is a common approach to address this challenge. However, in our specific case, this would result in an enlargement of the tube diameter. Moreover, it would necessitate an additional fabrication step, which may not be essential at the initial stage of validating the concept. This consideration is being taken into account for the future development of rolled 3D racetrack devices.

Page 11 line 2: the authors mention a significant H_c increase in the magnetic stripe upon rolling, but no results are shown to support this statement. Furthermore, the authors also assume an increase in PMA. A more convincing way should be used to support these claims.

Response: We appreciate the reviewer for noting this detail. The corresponding data, the change of the H_c , was added into the Supplementary section, as a figure 3. The reference to the Supp section was added to the main text. Concerning the PMA change: with our available instruments, we cannot make a direct estimation of the PMA change. However, we refer to other publications [49] with similar experiments, where the change of PMA under the strain was studied.

Also, could the authors comment on the role of magnetostriction of Co in these structures? Depending on the degree of bending and on the direction of applied stress, competing anisotropies (e.g. magnetoelastic, surface, dipolar) can lead to a change in the easy axis orientation. If such effects can be neglected in these stripes, the authors should comment why.

Response: We value the question raised by the reviewer; however, a more detailed investigation and discussion on this point are beyond the scope of our current research paper and are planned for follow-up studies. The answer on this question was given in this reply the text slightly above.

Page 19 line 12: Figures 1f and 1e are mentioned in the text but not available in the manuscript. Furthermore, additional schemes clarifying what is meant by tensile strain, rolling down, the positioning of the racetrack for Kerr microscopy analyses, could be added to facilitate visualization.

Response: We appreciate the reviewer for catching the typos. Indeed, there was a typo, and it has been corrected to Fig 1e. Additionally, black arrows have been incorporated in Figure 1c to denote the tensile strain, and corresponding text has been added to the captions. The rolling direction is depicted on Figure 1c. It's worth noting that the Kerr microscope is positioned above the sample, as illustrated in Figures 1a-c.

On Page 15 line 18 the authors say “The accuracy of this alignment is confirmed through hysteresis loop measurements. The Hall voltage is the same for an out of plane field of 0 Oe and 200 Oe ...”. This is an important step to ensure that the measurements are performed in comparable conditions and that no undesired components of the magnetic field are acting on the device. The authors are strongly encouraged to share these data/results with the readers.

Response: We concur with the reviewer that including this data will enhance the quality of the text. Accordingly, we have added the figure to the Supplementary section.

For figures 6b and 6d, the discussion indicates that the hysteresis loops the rolled structure “... exhibit a similar appearance compared to the planar case”. However, both the shape of the curves, width of the loop and its polarization are all strikingly different. This seem to indicate that quite different magnetic/electrical properties are present in each structure. Please comment or clarify this reasoning.

Response: Certainly, we agree with the reviewer's suggestion. As a result, this sentence has been removed from the text.

Again, results for more than one device result should be presented.

Response: We concur with the reviewer's perspective. Our presentation includes measurements conducted on at least two or more samples. In the text, we highlight the most representative results, which are indicative of the overall behavior observed across all devices. We have incorporated a comment on this aspect in the main text for clarity.

Page 16 line 7: the authors give the estimation of SOT efficiency but no framing within current most up-to-date results in the field are indicated especially for the planar structures which can be directly comparable.

Response: Thank you for your comment. We have addressed and corrected this section, incorporating additional references. You can find the revisions on page 17, line 5.

Page 18 line 16: the authors indicate that the temperature rise “does not boost the DW energy enough to overcome the additional pinning energy (barriers) arising from the strain induced PMA modification.” Have the authors considered other changes due to temperature rise that may also justify this behavior? For example, the use of highly temperature sensitive polymer substrates which may have implications in different strains as a function of temperature, or the change of effective anisotropy as a function of temperature. Could the authors comment if this or similar effects may be relevant for this study and envisaged final application, or if not why.

Response: Thank you for your comment. Indeed, this is a valid technical concern. We have conducted a rough estimation of the thermally induced PI expansion for our 3D RT. In this simplified model, we assumed that the PI would be locally heated (under the ferromagnetic stripe) during the application of the current pulse. Upon turning off the current pulse, heat dissipation from the PI/RT interface is rapid, and thus, we only consider the shape change during the current pulse.

The thermal expansion coefficient was obtained from the data sheet of the Kapton film, which is chemically similar to our polyimide material (<https://www.dupont.com/content/dam/dupont/amer/us/en/ei-transformation/public/documents/en/K-15361-Kapton-FPC-DataSheet.pdf>). We adopted a thermal coefficient of expansion equal to $17 \times 10^{-6} \text{ K}^{-1}$, considering the maximum operational temperature difference for the 3D racetrack (as indicated in Figure 8d) to be 76 K. The resulting thermally induced strain is calculated to be 0.12%, which accounts for 9% of the geometry-induced strain. We consider this value as a top estimation. It is important to note that metallic contacts can function as a thermal bath, facilitating the dissipation of Joule heat and mitigating the overall thermally induced strain effect. In conclusion, we anticipate that the contribution will be sufficiently small to be negligible.

The authors then proceed to characterize the DMI field via bubble expansion technique imaged by Kerr microscopy. The authors should clearly identify in Figure 9 and in the text if such studies are done in a planar geometry or rolled structure for the polymer substrate.

Response: We appreciate the reviewer for bringing this comment to our attention. The omission has been rectified.

Although studies in full planar structures grow on polyimide can help address some of the underlying roots of the decrease in the DW velocity and support the assessment of DMI field, the real impact of all variables and the most relevant measurement would come from 3D rolled structures being in the same conditions as the final device. In the planar polymer there is no strain from bending (only intrinsic) and no induced cracks.

In page 20 line 20 the authors then state that they “anticipate that the interfacial DMI will remain relatively unchanged in the 3D state under 1.3% strain.” This statement is not clearly substantiated. Once under strain it is not clear how several factors such as surface anisotropy, additional defects, surface changes, etc, are expected to stay the same. The authors should elaborate, clarify, or support further with literature their statement. Ref 48 is a start but the differences in the fabrications methods, eg substrate surface characteristics, critical interfaces with Pt and Ir, or the size of the measured structures are quite different.

Response: We appreciate the reviewer for this comment. Throughout the rolling process, we do not anticipate any modification to the FM surface or material due to the influence of chemical compounds. The FM layer is shielded by Pt and Ta layers, known for their low chemical interactions. Additionally, we introduced benzotriazole as a corrosion inhibitor to encapsulate metallic structures, thereby preventing surface modification. Investigating changes in parameters such as surface anisotropy, additional defects, and surface alterations would require a dedicated research effort beyond the scope of our current manuscript. Furthermore, the DMI undergoes variations in nontrivial geometries, as reported by Robert Streubel et al. in "Magnetism in curved geometries" (*J. Phys. D: Appl. Phys.* 49, 363001, 2016). The curvature effect becomes significant in structures with a curvature diameter of tens of nanometers. In our case, the tube diameter is 32 μm , leading us to assume that the DMI does not exhibit considerable variation. We have incorporated additional remarks on these points into both the main text and outlook section for clarification.

The use of CGS and SI units through the manuscript should be revised. The authors indicate several times H in Oe and $\mu_0 H$ in Oe. Especially in the graphics there is no consistency in labeling the magnetic field (e.g. figure 6F and figure 6D). Systematic use of SI units for magnetic field should be implemented.

Response: We appreciate the reviewer for bringing this important oversight to our attention. The necessary changes have been incorporated into the manuscript.

Materials and Methods

- Page 27 line 17: the authors mentioned the spin-coating of a discharging layer for the EBL process. The materials, or types of polymers used should be indicated.
- Page 28 lines 12-17: No mentioning of baking steps are given for the polymers in particular regarding polyimide. It is quite well known that different curing temperatures can be used, but depends significantly on the type of polyimide that is being employed. This information should be included. Also, the hydrogel is said to be spin-coated with a trapezoidal shape. Maybe such shape is obtained by subsequent lithography. If so the authors should elaborate on which steps of mask definition and pattern transfer conditions are used. These are important information for the work to be reproduced.

Minor typos:

- page 3 line 1: “has been examined in films”
- page 3 line 21: “velocities at the angular momentum compensation temperature”
- page 13 line 13: “as a function of I_{DC} ”
- page 27 line 14: units of pressure are missing
- page 20 line 17” units of fields are missing in H_{DMI}

Thank you for paying attention to such details and fishing out these deficiencies. We have implemented the abovementioned corrections accordingly.

REVIEWERS' COMMENTS

Reviewer #1 (Remarks to the Author):

I would like to express my gratitude to the authors for their hard work in improving the manuscript and addressing my review. I am confident that the manuscript, in its current form, meets the journal's requirements and will be of interest to wide readers. The authors' responses to my queries and those of the other reviewers are persuasive. Therefore, I recommend the manuscript for publication in its current form.

Reviewer #2 (Remarks to the Author):

I appreciate the authors' efforts in answering my questions. All my previous concerns have been adequately addressed by the authors, and their manuscript has greatly improved. Thus, I recommend their publication in Nature Communications.

Reviewer #3 (Remarks to the Author):

The authors have addressed all my questions in a very satisfactory manner. The elaboration on the potential underlying effects that justify the observed behaviors is highly appreciated. Also, I would like to thank the discussions and justifications provided throughout the reply to the reviewers, and the follow up thorough revisions of their manuscript. The inclusion of several experimental details, addition of extra data/results on the supplementary material, and on the rebuttal has increased the robustness and credibility of the work. The title change also reflects very well the scope of the work. I recommend the work for publication.